# Low FoxO expression in *Drosophila* somatosensory neurons protects dendrite growth under nutrient restriction

Amy R Poe[†‡], Yineng Xu[†], Christine Zhang, Joyce Lei, Kailyn Li, David Labib, Chun Han*

Weill Institute for Cell and Molecular Biology and Department of Molecular Biology and Genetics, Cornell University, Ithaca, United States

**Abstract** During prolonged nutrient restriction, developing animals redistribute vital nutrients to favor brain growth at the expense of other organs. In *Drosophila*, such brain sparing relies on a glia-derived growth factor to sustain proliferation of neural stem cells. However, whether other aspects of neural development are also spared under nutrient restriction is unknown. Here we show that dynamically growing somatosensory neurons in the *Drosophila* peripheral nervous system exhibit organ sparing at the level of arbor growth: Under nutrient stress, sensory dendrites preferentially grow as compared to neighboring non-neural tissues, resulting in dendrite overgrowth. These neurons express lower levels of the stress sensor FoxO than neighboring epidermal cells, and hence exhibit no marked induction of autophagy and a milder suppression of Tor signaling under nutrient stress. Preferential dendrite growth allows for heightened animal responses to sensory stimuli, indicative of a potential survival advantage under environmental challenges.

**\*For correspondence:**
chun.han@cornell.edu

[†]These authors contributed equally to this work

**Present address:** [‡]Department of Psychiatry; Chronobiology and Sleep Institute, Perelman School of Medicine, Universityof Pennsylvania, Philadelphia, United States

**Competing interests:** The authors declare that no competing interests exist.

## Introduction

Proper animal development requires coordinated growth of various organs to achieve the correct relative organ proportions in mature individuals. However, when developing animals face adverse conditions, such as limited availability of nutrients, they reallocate essential resources to favor growth of vital organs at the expense of other organs. This phenomenon of 'organ sparing' is exemplified by the preferential growth of the brain in human fetuses experiencing intrauterine growth restriction, resulting in undersized newborns with disproportionately large heads (*Gruenwald, 1963*). Although changes of circulation are known to enhance oxygen and blood supply towards the brain under nutrient deprivation in mammals (*Severi et al., 2000*; *Cohen et al., 2015*), how systemic growth control is altered at the molecular level to favor brain development remains poorly understood.

A well-characterized example of brain sparing occurs in *Drosophila* larvae experiencing nutrient deprivation. Systemic larval body growth of *Drosophila* is controlled by the conserved insulin/insulin like growth factor (IGF) pathway (*Rulifson et al., 2002*). *Drosophila* insulin-like peptides (Dilps) secreted by the insulin producing cells (IPCs) in the larval brain promote cell proliferation and growth of peripheral tissues by activating the insulin receptor (InR) and the downstream signaling components phosphatidylinositol 3-kinase (PI3K) and Akt (PKB) (*Verdu et al., 1999*; *Brogiolo et al., 2001*; *Ikeya et al., 2002*; *Oldham et al., 2002*). Nutrient restriction suppresses insulin secretion through an intricate nutrient sensing mechanism involving inter-organ communications between the fat body and IPCs (*Ikeya et al., 2002*; *Géminard et al., 2009*; *Rajan and Perrimon, 2012*), and consequently, curbs the growth of most peripheral tissues. However, the larval brain is protected against nutrient deprivation and exhibits continuous neurogenesis (*Cheng et al., 2011*). This protection is mediated

**eLife digest** The organs of a young animal develop in a carefully controlled way to reach the right size relative to each other. However, if the animal's diet does not contain the right amount of nutrients — a condition known as malnutrition – the body prioritizes the needs of the brain and other vital organs. This means that certain organs keep on growing while others stop.

The brain is at the center of the nervous system, which is formed of networks of nerve cells (or neurons) that rapidly carry messages around the body. In the larvae of malnourished fruit flies, a molecular signal allows the nervous system to continue making new neurons as other parts of the body slow down their growth.

During development, neurons also connect to each other by growing tree-like structures known as dendrites. However, it remained unclear whether the growth of dendrites was also protected during episodes of malnutrition.

To address this question, Poe, Xu et al. performed experiments in the larvae of fruit flies, focusing on a type of neuron whose dendrites extend into the skin. When nutrients were scarce, the neurons grew more rapidly than the surrounding skin cells, resulting in dendrite overgrowth. Compared to neurons, the skin cells had higher levels of a stress sensor known as FoxO, which stops cell growth when nutrients are scarce. Conversely, low quantities of FoxO in neurons allow these cells to keep on growing dendrites, which ultimately helps the starved animals to better react to their environment.

These results suggest that the growth of neurons and their connecting structures is preserved during malnutrition. Ultimately, dissecting how organisms prioritize resources can help to develop new approaches to treat human conditions that emerge during malnutrition.

by the glia-derived Jelly belly (Jeb) ligand that activates the Anaplastic lymphoma kinase (Alk) receptor on neural stem cells (NSCs) to turn on the downstream PI3K pathway independent of nutrition (*Cheng et al., 2011*). Although cell proliferation of the nervous system is spared under nutrient deprivation, whether other aspects of neural development are also subject to organ sparing is unknown.

The arbor growth of post-mitotic neurons is achieved by cell expansion rather than cell number increase and therefore represents a different type of neural growth from cell proliferation. Following innervation of the target field, the dendritic or axonal arbor of the neuron expands in coordination with the tissue it innervates. For example, the dendritic arbors of *Drosophila* somatosensory neurons called dendritic arborization (da) neurons are known to scale with the body wall during normal larval development (*Parrish et al., 2009*). This scaling involves synchronous expansion of body wall epidermal cells and of da dendritic arbors, such that neurons maintain the same coverage of the sensory fields while the body surface area expands exponentially (*Jiang et al., 2014*). Da neurons are categorized into four classes that differ in their dendrite morphology and transcription factor expression (*Grueber et al., 2002*; *Hattori et al., 2013*). Recently, class IV da (C4da) neurons, which completely cover the body surface and thus are called 'space-filling' neurons (*Grueber et al., 2002*; *Grueber et al., 2003*), were found to elaborate more dendrite branches when larvae develop on a low-nutrient diet (*Watanabe et al., 2017*), suggesting that dendritic scaling of C4da neurons is regulated by the nutrient state. However, whether this dendritic hyperarborization is related to organ sparing and how nutrient stress promotes dendrite growth are unclear.

The conserved PI3K-Akt-mechanistic target of rapamycin (mTOR) pathway promotes dendrite growth in both insects and mammals (*Jaworski et al., 2005*; *Kumar et al., 2005*; *Parrish et al., 2009*; *Skalecka et al., 2016*). Receiving signaling inputs from membrane receptor tyrosine kinases (RTKs), notably InR (*Sancak et al., 2007*; *Vander Haar et al., 2007*; *Wang et al., 2007*), this pathway enhances translation in most cells by mTOR kinase-mediated phosphorylation of S6 protein kinase (S6K) and 4E-binding protein (4E-BP) (*Burnett et al., 1998*). At the center of this pathway, mTOR activity is also influenced by the cellular state, including nutrient availability, cellular energy levels, and stress factors (*Zoncu et al., 2011*). In particular, cellular nutrient starvation suppresses mTOR and consequently induces autophagy (*Ganley et al., 2009*; *Hosokawa et al., 2009*; *Jung et al., 2009*), the self-eating process that helps to conserve and recycle vital cellular building blocks. mTOR regulates autophagy in part through the transcription factor EB (TFEB), which promotes

autophagosome biogenesis but is suppressed by mTOR-mediated phosphorylation (*Jung et al., 2009*; *Martina et al., 2012*; *Roczniak-Ferguson et al., 2012*). Among the cellular stress sensors that inhibit mTOR activity, the forkhead box O (FoxO) family of transcription factors can be activated by a variety of stress signals and respond by suppressing cell growth and inducing autophagy (*Eijkelenboom and Burgering, 2013*). Although the regulation of mTOR activity by cellular stress has been extensively investigated in many cell types, how mTOR signaling is modulated by the nutrient state to impact neuronal arbor growth has not been examined. Furthermore, although FoxO members have been found to enhance dendritic space-filling of C4da neurons in *Drosophila* (*Sears and Broihier, 2016*) and to regulate dendrite branching and spine morphology of adult-generated neurons in mice (*Schäffner et al., 2018*), whether they also influence neuronal arbor growth in response to nutrient stress is unclear.

In this study, we demonstrate that dynamically growing *Drosophila* da neurons exhibit organ sparing at the level of individual cells, with dendrites growing preferentially at the expense of other non-neural tissues under nutrient stress. Mechanistically, the amplitude of Tor signaling in da neurons is attenuated less dramatically by nutrient stress than in non-neural tissues like epidermal cells, muting the induction of autophagy in neurons. The distinct sensitivities of da neurons and epidermal cells to nutrient stress are at least partly due to their differential FoxO expression levels: Foxo is lowly expressed in neurons and hence has minimal effect on dendrite growth, while it is highly expressed in epidermal cells, resulting in suppression of cell growth only under nutrient restriction. Functionally, preferential dendrite growth of da neurons increases the sensory acuity, allowing larvae to respond more nimbly to environmental stimuli.

## Results

### Nutrient restriction affects the growth of epidermal cells and C4da neurons differentially

A recent study by *Watanabe et al., 2017* reported that C4da neurons were hyperarborized when *Drosophila* larvae developed on a low yeast diet that restricts the availability of lipids and amino acids, an interesting and surprising finding that agrees with our independent observation. In our experiments, we examined larvae reared in high yeast (HY, 8% yeast) and low yeast (LY, 1% yeast) media that otherwise contained only glucose as a carbon source. In wandering third instar larvae, C4da neurons in the HY condition showed sparse dendrites with gaps of dendritic coverage between neighboring neurons (*Figure 1A*). In contrast, C4da neurons in the LY condition completely covered the body wall with dense dendrites (*Figure 1B*). This apparent dendrite overgrowth in the LY condition (*Figure 1D*) was due to 60% more total branches and 59% more terminal branches but not increase of terminal branch length as compared to the HY condition (*Figure 1—figure supplement 1*).

The increased density of epidermal innervation by C4da dendrites suggests that nutrient restriction differentially affected growth of C4da neurons and the body wall. To investigate this possibility, we monitored growth of C4da neurons and the body wall simultaneously during larval development in HY and LY media. We measured various parameters of the larval body wall and dendrite growth every 24 hr (hrs) starting from 48 hr after egg laying (AEL) to the wandering 3rd instar stage (*Figure 1* and *Figure 1—figure supplement 2*). The body wall was measured at the levels of the whole body (body length), the body segments (segment width), and individual epidermal cells (average cell width as visualized by the septate junction marker Nrg-GFP). We found that all three parameters correlated with one another in both HY and LY conditions (*Figure 1—figure supplement 2G and H*). We have therefore taken the segment width as an indicator of the larval body size.

Compared to the HY condition, LY caused a significant delay in larval body growth, with animals in LY taking 2.2 times longer (264 hr compared to 120 hr) to pupariate. Notably, larvae reared in HY and LY reached a nearly identical maximum segment width of ~600 μm before pupariation (*Figure 1A–C*). These observations verify that the LY medium caused the larvae to experience nutrient stress and developmental delay. The fact that the pupariation occurs at the same maximum segment width regardless of the rate of growth suggests that the segment width is not only a good proxy for body size, but it also provides a good indication of the developmental stage.

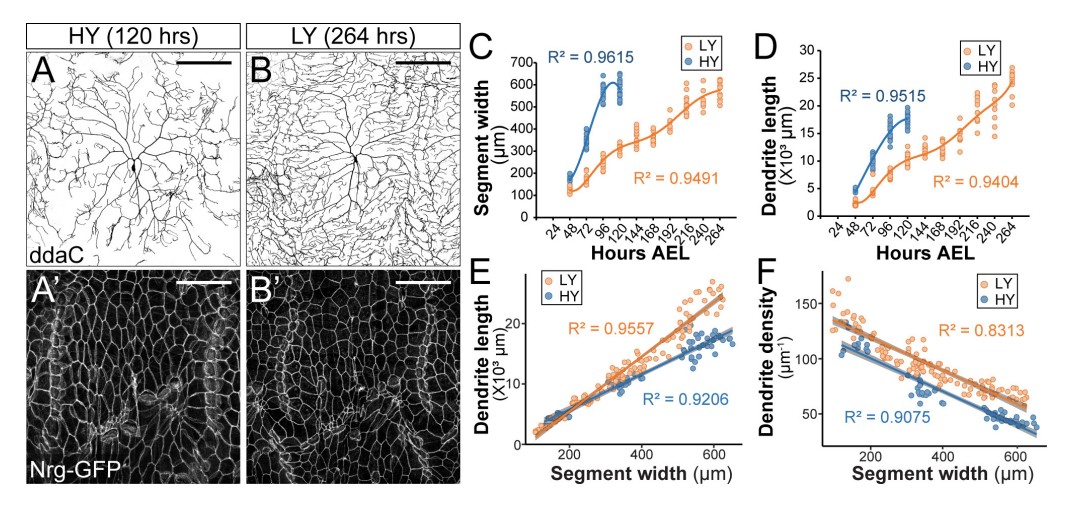

**Figure 1.** Nutrient restriction affects the growth of epidermal cells and C4da neurons differentially. (A–B′) Double labeling of ddaC neurons by *ppk-CD4-tdTom* (A and B) and epidermal cells by the septate junction marker Nrg-GFP (A′ and B′) in the high yeast (HY, 8%) condition at 120 hr after egg laying (AEL) (A and A′) and in the low yeast (LY, 1%) condition at 264 hr AEL (B and B′). (C and D) Plots of segment width (C) and total dendrite length of ddaC neurons (D) versus time in HY and LY conditions. (E and F) Plots of total dendrite length ($p \ll 0.05$) (E) and dendrite density (total dendrite length/dendrite coverage area, $p \ll 0.05$) (F) with segment width in HY and LY conditions. Each circle represents a segment in (C) and a ddaC neuron in (D–F); n = 63 for HY; n = 115 for LY. Solid lines represent polynomial fits in (C) and (D) and linear fits in (E) and (F). $R^2$ represents coefficient of determination of the linear regression. Gray shading in (E) and (F) represents a 0.95 confidence interval (CI) of the linear model. *p*-value represents the possibility that the slopes of two yeast conditions are the same. Scale bars, 100 µm.

The online version of this article includes the following source data and figure supplement(s) for figure 1:

**Source data 1.** Larvae growth data for *Figure 1* and *Figure 1—figure supplement 1*, *2*, *3*.
**Figure supplement 1.** The effects of nutrient restriction on dendrite growth of C4da neurons.
**Figure supplement 2.** Development of C4da neurons and epidermal cells in high and low yeast.
**Figure supplement 3.** Growth of the larval body and C4da neurons under starvation.

C4da neurons also grew slower in LY than in HY, as indicated by smaller increases of the total dendrite length during each 24 hr period (*Figure 1D*). However, because LY larvae had more time to develop, their neurons ultimately outgrew those of HY larvae, resulting in 1.39-fold longer total dendrites at the end of the larval period (*Figure 1D*). To make more meaningful comparisons of dendrite growth in animals of similar developmental stages prior to pupariation, we plotted the dendrite length against the segment width (*Figure 1E*), as the segment width better indicates the developmental stage than the age of the larva. This plot shows that C4da dendrites grow 57% faster (based on the slopes of the linear fits) in LY than in HY when normalized by the segment width. Interestingly, the dendrite density (dendrite length/dendrite field size) was always higher in LY when plotted against the segment width (*Figure 1F*), even though the dendrite length was not greater in LY when the segment width was below 200 µm (*Figure 1E*). Likely contributing to this discrepancy between dendrite density and dendrite length is that larvae reared in LY were thinner than those in HY (*Figure 1—figure supplement 2I and J*) and consequently had smaller body wall areas to be covered by C4da dendrites.

The above data suggest that C4da neurons preferentially grow compared to epidermal cells when nutrients are limited. To determine if C4da neurons still have a growth advantage under starvation, we transferred larvae reared in HY media to agar-only media at 84 hr AEL, a time when larvae had reached the critical weight (*Beadle et al., 1938*; *Bakker, 1959*). Interestingly, C4da neurons showed significant dendrite growth as measured by the dendrite length normalized to the segment width (abbreviated as normalized dendrite length) over a 24 hr period (*Figure 1—figure supplement 3A*), even though the larval body size did not change in the same period (*Figure 1—figure supplement 3B*). Collectively, these results show that while nutrient restriction delays overall larval growth, it differentially affects growth of C4da neurons and epidermal cells, such that C4da neurons exhibit a growth advantage, or are spared, under nutrient stress. Moreover, in the absence of

nutrient intake, larvae mobilize existing nutrient storage to support the growth of C4da neurons at the expense of non-neural tissues like epidermal cells.

## The InR-Tor pathway mediates the preferential dendrite growth under nutrient stress

Nutrient-mediated systemic control of larval growth depends on InR and the *Drosophila* mTOR homolog Tor (*Boulan et al., 2015*). To examine effects of the InR/Tor pathway on dendrite growth during nutrient stress, we selectively inactivated InR or Tor in C4da neurons. For these assays, we measured normalized dendrite length when larvae reached a segment width of 500–550 µm, a size at which C4da neurons in control larvae exhibit a significant increase in dendrite growth in response to nutrient stress (*Figure 1E*). Downregulation of the InR/Tor pathway in neurons by *InR* knockdown (*InR* RNAi), *InR^{DN}* (dominant negative) overexpression (*InR* DN), *Tor* knockdown (*Tor* RNAi), or *Tor^{DN}* overexpression (*Tor* DN) in the HY condition caused mild or statistically non-significant dendrite reduction compared to the control (*Figure 2A–C and G*, and *Figure 2—figure supplement 1A–1C*). In contrast, the same genetic manipulations caused pronounced dendritic reduction in the LY condition, generating dendrite patterns resembling those of wildtype neurons in the HY condition (*Figure 2D–F and G*, and *Figure 2—figure supplement 1D–1F*). Statistical analyses suggest that the effects of the manipulations are nutrient-dependent, that is, much greater reduction of normalized dendrite length in the LY condition. The ratio of the average normalized dendrite length between LY and HY thus drops from 1.44 for control neurons to values closer to one for neurons in which *InR* or *Tor* is suppressed (*Figure 2H*). These results suggest that neuronal InR/Tor signaling is responsible for the preferential dendrite growth observed under nutrient stress.

To test whether reducing the rate of epidermal growth, as is seen under nutrient stress, can lead to excessive dendrite growth, we inhibited *InR* and *Tor* in epidermal cells under the HY condition. The efficacy of genetic manipulations in attenuating epidermal growth was assessed using *Gal4^{R16D01}*, which is expressed in a stripe of epidermal cells in the middle of each segment (*Poe et al., 2017*; *Figure 2—figure supplement 1G*), allowing comparison of Gal4-expressing epidermal cells to neighboring wildtype cells. *UAS*-driven transgenes were then expressed by a pan-epidermal driver *Gal4^{R38F11}* (*Figure 2—figure supplement 1H*) to inhibit *InR* or *Tor* in the entire epidermal sheet. Downregulating *InR* or *Tor* effectively reduced the epidermal cell size as expected (*Figure 2—figure supplement 1I–1M*), reducing the ratio between the sizes of Gal4-positive (Gal4) and neighboring Gal4-negative (WT) cells (*Figure 2—figure supplement 1N*). Suppressing *InR* or *Tor* function throughout the epidermal sheet delayed epidermal growth such that larvae took 6–30 extra hours to reach the segment width of 500–550 µm, at which size C4da neurons exhibited 22–61% greater normalized dendrite length than controls (*Figure 2I–L* and *Figure 2—figure supplement 1O–1Q*). Taken together, the above results demonstrate that changes in the relative strengths of the InR/Tor signaling in neurons and epidermal cells can alter dendritic scaling: Reducing the throughput of the pathway in neurons can cause dendrite reduction, while suppression of InR/Tor in the epidermis can lead to dendrite overgrowth.

To determine how nutrient levels modulate InR/Tor signaling in neurons and epidermal cells, we examined Tor activity by immunostaining phosphorylated Ribosomal protein S6 (pRpS6), a substrate of S6K and a faithful indicator of Tor activity in *Drosophila* tissues (*Ruvinsky and Meyuhas, 2006*; *Kim et al., 2017*). In HY, the cytoplasm of epidermal cells exhibited high and uniformly distributed pRpS6 signals, while the soma and primary dendrites of C4da neurons showed comparatively low pRpS6 intensities (*Figure 2M*). As a result, the ratios of pRpS6 intensity (average intensity within regions of interest) between the neuronal compartments and epidermal cells are much lower than 1 (*Figure 2P*). Under nutrient stress, the overall pRpS6 staining on the larval body wall dramatically decreased (*Figure 2O*), consistent with the notion that nutrient stress reduces InR/Tor signaling in peripheral tissues (*Géminard et al., 2009*). However, in these animals, the pRpS6 signals were brighter and more even in C4da cell bodies and dendrites than in the epidermal cells (*Figure 2N*), causing the ratios of pRpS6 intensity between the neuronal compartments and epidermal cells to be larger than 1 (*Figure 2P*). These data suggest that nutrient stress switches the relative strength of the InR/Tor signaling in C4da neurons and epidermal cells such that the neurons gain a growth advantage over epidermal cells.

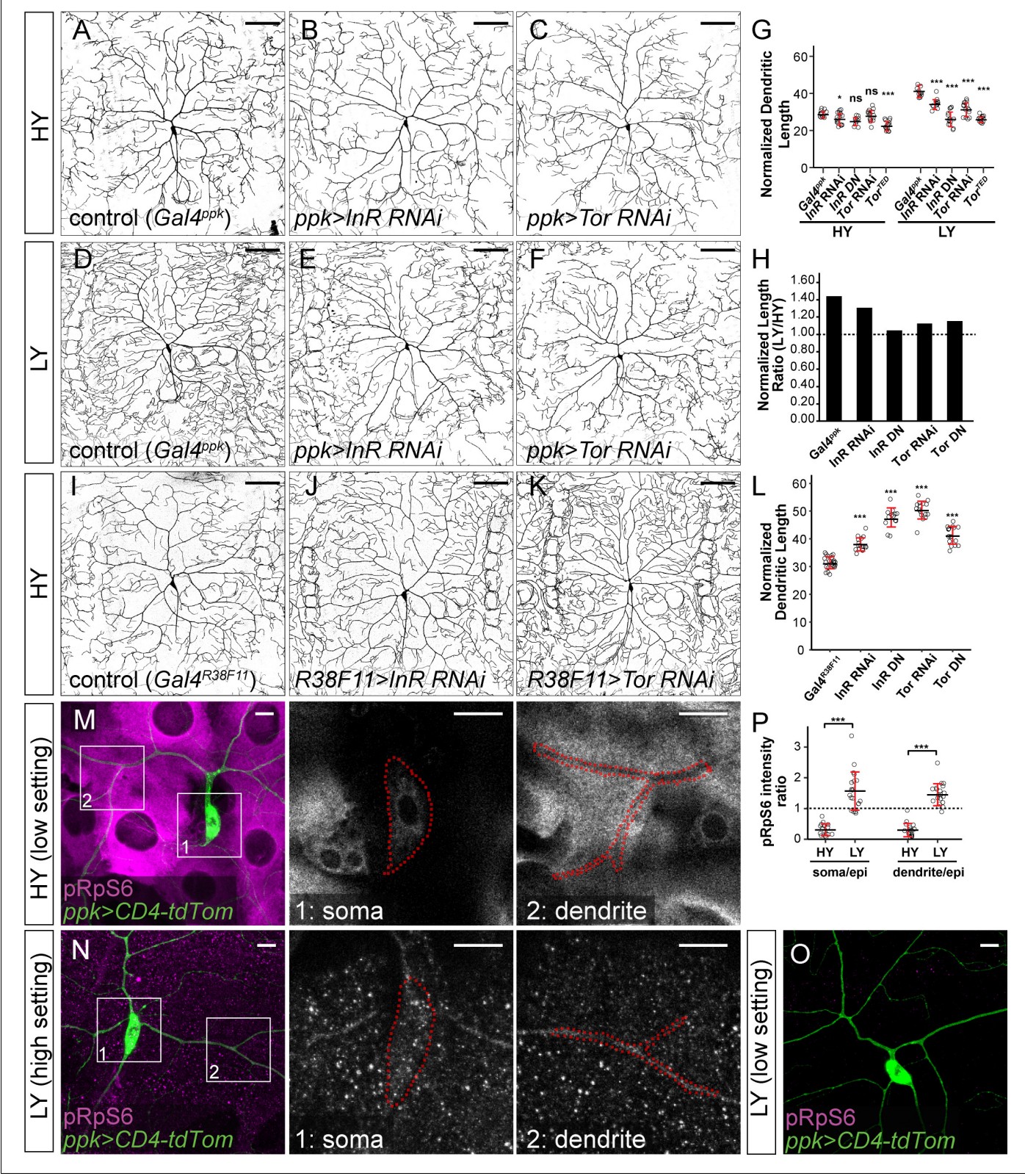

**Figure 2.** The InR-Tor pathway underlies the preferential neuronal growth under nutrient stress. (**A–F**) ddaC neurons in the *Gal4^ppk* control (**A and D**) and animals expressing *Gal4^ppk*-driven *InR* RNAi (**B and E**) and *Tor* RNAi (**C and F**) in HY and LY conditions. (**G**) Quantification of normalized dendrite length (total dendrite length/segment width) in HY and LY conditions. HY: n = 14 for *Gal4^ppk*, n = 15 for *InR* RNAi, n = 11 for *InR* DN, n = 15 for *Tor* RNAi, n = 15 for *Tor* DN; LY: n = 14 for *Gal4^ppk*, n = 12 for *InR* RNAi, n = 14 for *InR* DN, n = 15 for *Tor* RNAi, n = 15 for *Tor* DN. Two-way ANOVA,

*Figure 2 continued on next page*

*Figure 2 continued*

Posthoc contrasts with a Dunnett correction. The differences between control and *InR* RNAi under HY and LY conditions are significantly different as indicated by a significant interaction term (p=0.003924), the same for *InR* DN (p=8.581e-09), *Tor* RNAi (p=5.876e-07) and *Tor* DN (p=1.062e-09). (H) The ratios of average normalized dendrite length between LY and HY. (I–K) ddaC neurons in the *Gal4^R38F11* control (I) and animals expressing *Gal4^R38F11*-driven *InR* RNAi (J) and *Tor* RNAi (K) in HY condition. (L) Quantification of normalized dendrite length in e *Gal4^R38F11*-driven knockdown and overexpression. n = 23 for *Gal4^R38F11*, n = 17 for *InR* RNAi, n = 17 for *InR* DN, n = 16 for *Tor* RNAi, n = 17 for *Tor* DN. One-way ANOVA and Tukey's HSD test. (M–O) pRpS6 staining (magenta) of ddaC neurons (Green) and epidermal cells in HY and LY conditions in 2-dimensional (2D) projections. The insets in (M) and (N) show pRpS6 staining at the soma (1) and primary dendrites (2) in single confocal sections, with the somas and dendrites outlined. High settings and low settings stand for high and low pRpS6 detection settings. (P) Quantification of pRpS6 intensity ratio (soma/epidermal cells and dendrites/epidermal cells) in HY and LY conditions. Soma/epi: n = 17 for HY, n = 20 for LY; dendrites/epi: n = 16 for HY, n = 20 for LY. Two-way ANOVA. The differences between HY and LY in soma and dendrites are not significantly different as indicated by a non-significant interaction term (p=0.5592). For all quantifications, ***p<0.001; *p<0.05; ns, not significant. each circle represents a neuron. Significance level is for comparison between the control and the genotype indicated under the same food condition. Black bars, mean; red bars, SD. Scale bars, 100 µm in (A–K); 10 µm in (M–O). The online version of this article includes the following source data and figure supplement(s) for figure 2:

**Source data 1.** InR-Tor pathway manipulation data for *Figure 2* and *Figure 2—figure supplement 1*.
**Figure supplement 1.** The effects of suppressing *InR* and *Tor* in ddaC neurons and epidermal cells in HY and LY conditions.

## The lack of autophagy induction protects C4da neuronal growth under nutrient stress

Nutrient stress suppresses mTOR activity to induce many cellular responses, including autophagy (*He and Klionsky, 2009*). We therefore tested if autophagy is also differentially regulated by nutrient levels in C4da neurons and epidermal cells. To monitor autophagy levels, we used an mCherry-Atg8a reporter under the control of the endogenous Atg8a regulatory sequence, which labels autophagic structures (*Hegedűs et al., 2016*). As expected, autophagosome levels in epidermal cells were low under the HY condition (*Figure 3A*) but increased nearly 5-fold in the LY diet (*Figure 3B and C*). In contrast, autophagosomes in C4da cell bodies were present at low levels under both HY and LY conditions (*Figure 3A–C*). Similarly, epidermal expression of Lamp-mCherry, a lysosomal and autolysosomal marker driven by the endogenous *Lamp1* promoter (*Hegedűs et al., 2016*), increased by 4.8-fold under nutrient stress (*Figure 3—figure supplement 1A–1C*). In contrast, the same reporter exhibited a much lower baseline labeling in C4da cell bodies (12.6% of that in epidermal cells) under HY and non-significant increase under LY (*Figure 3—figure supplement 1A–1C*). To examine the levels of autophagic flux, we overexpressed in epidermal cells and neurons a tandem fluorescent marker GFP-mCherry-Atg8a, which is converted from GFP + mCherry dual fluorescence to mCherry alone when autophagosomes mature (*Kimura et al., 2007*; *Nezis et al., 2010*). Thus, increased autophagic flux results in a reduction of cytosolic GFP and an increase of vesicular mCherry, indicated by an increased ratio of mCherry area over GFP intensity. Using this marker, we found that although nutrient stress enhanced autophagic flux in both epidermal cells and C4da neuron cell bodies, the autophagic flux level was always higher in epidermal cells (*Figure 3—figure supplement 1D–1I*). These data suggest that C4da neurons maintain low levels of autophagy even under nutrient stress, despite an increase of autophagic flux.

To examine the effects of autophagy on the growth of epidermal cells and neurons, we knocked down *Atg8a* to suppress autophagy. Epidermal *Atg8a* knockdown using *Gal4^R16D01* had no effect on cell size under the HY condition (*Figures 3D, E and H*), consistent with the low autophagy level in these cells. However, the same manipulation caused a 16% increase of the epidermal cell size under LY (*Figure 3F and G*, and 3H), supporting the idea that autophagy suppresses epidermal cell growth under nutrient stress. In comparison, neuronal *Atg8a* knockdown in both HY and LY conditions resulted in similar increases of normalized dendrite length (13% and 11% increases, respectively) (*Figure 3I* and *Figure 3—figure supplement 2A–2D*), suggesting that C4da neurons maintain a constant level of autophagy regardless of nutrient availability to mildly suppresses dendrite growth.

Nutrient restriction upregulates expression of autophagy-related genes through the transcription factor TFEB, a substrate of the Tor kinase (*Füllgrabe et al., 2016*). To understand why nutrient stress fails to induce autophagy in neurons, we examine the expression of *Mitf-GFPnls*, a transcription reporter for the *Drosophila* TFEB homolog *Mitf* (*Zhang et al., 2015*). *Mitf-GFPnls* showed nutrient-independent expression in epidermal nuclei but its expression could not be detected in da neurons in either HY or LY condition (*Figure 3—figure supplement 2E–G*), suggesting that Mitf transcription

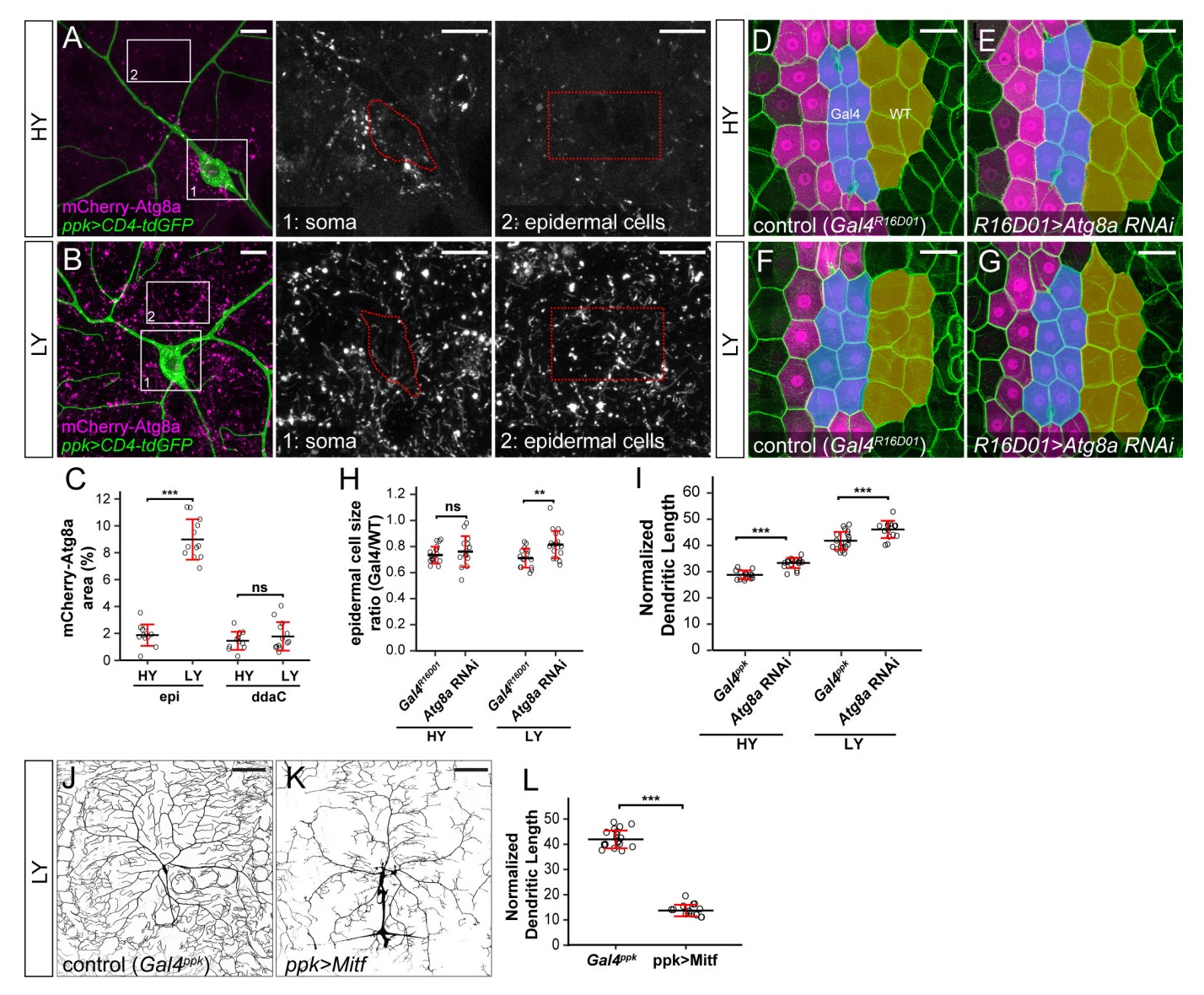

**Figure 3.** The lack of autophagy induction protects ddaC neuron growth under nutrient stress. (**A and B**) mCherry-Atg8a (magenta) in ddaC soma (green) and epidermal cells in HY and LY conditions. The insets show mCherry-Atg8a at the soma of ddaC (1) and epidermal cells (2). The dotted lines indicate the somas (1) and measured epidermal regions (2). All images are 2D projections. (**C**) Quantification of mCherry-Atg8a levels in ddaC soma and epidermal cells in HY and LY conditions, measured by the area percentage of mCherry-Atg8a-positive vesicles. Epi: n = 12 for HY, n = 13 for LY; ddaC: n = 12 for HY, n = 13 for LY. Two-way ANOVA. The differences between HY and LY in epidermal cells and ddaC are significantly different as indicated by a significant interaction term (p=2.416e-14). (**D–G**) Epidermal cells in the *Gal4^{R16D01}* control and animals expressing *Gal4^{R16D01}*-driven *Atg8a* RNAi in HY and LY conditions. *Gal4^{R16D01}* domain is labeled by mIFP expression (magenta). All epidermal cells are labeled by Nrg-GFP (green). The blue and yellow overlays indicate the measured Gal4-expressing and wildtype (WT) epidermal cells, respectively. (**D**) is the same as *Figure 2— figure supplement 1I*. (**H**) Quantification of epidermal cell size ratio (*Gal4^{R16D01}* cells/WT cells) in HY and LY conditions. Each circle represents a segment; HY: n = 18 for control, n = 14 for *Atg8a* RNAi; LY: n = 16 for control, n = 19 for *Atg8a* RNAi. Two-way ANOVA. (**I**) Quantification of normalized dendritic length in control and *Atg8a* RNAi animals in HY and LY conditions. Each circle represents a neuron; HY: n = 14 for *Gal4^{ppk}*, n = 16 for *Atg8a* RNAi; LY: n = 14 for *Gal4^{ppk}*, n = 12 for *Atg8a* RNAi. *Gal4^{ppk}* is the same dataset as in *Figure 2G*. Two-way ANOVA. The differences between control and *Atg8a* RNAi under HY and LY are not significantly different as indicated by a non-significant interaction term (p=0.648) (**J and K**) DdaC neurons in the *Gal4^{ppk}* control (**J**) and animals expressing *Gal4^{ppk}*-driven *Mitf* (**K**) in LY condition. (**J**) is the same as *Figure 2D*. (**L**) Quantification of normalized dendritic length in control and *Mitf* overexpression animals in LY condition. Each circle represents a neuron; n = 14 for *Gal4^{ppk}*, n = 15 for *ppk >Mitf*. *Gal4^{ppk}* is the same dataset as in *Figure 2G*. Student's t-test. For all quantifications, ***p<0.001; **p<0.01; ns, not significant. Black bars, mean; red bars, SD. Scale bars, 10 μm in (**A**) and (**B**); 50 μm in (**D–G**); 100 μm in (**J**) and (**K**).

The online version of this article includes the following source data and figure supplement(s) for figure 3:

*Figure 3 continued on next page*

*Figure 3 continued*

**Source data 1.** Autophagy data for *Figure 3* and *Figure 3—figure supplements 1*, *2*.
**Figure supplement 1.** Autophagy flux in C4da neurons and epidermal cells.
**Figure supplement 2.** Effects of *Atg8a* knockdown in C4da neurons and *Mitf-nGFP* expression pattern.

may be low in da neurons. We next investigated the effects of inducing autophagy on dendrite growth by overexpressing Mitf in C4da neurons, as overexpression of TFEB/Mitf is sufficient to dominantly induce autophagy in cells (*Settembre et al., 2011*; *Zhang et al., 2015*). This manipulation strongly reduced the normalized dendrite length under LY (*Figure 3J–L*), suggesting that high autophagy levels suppress dendrite growth.

These data together suggest that nutrient restriction upregulates autophagy in epidermal cells but not in C4da neurons, and that the lack of autophagy induction likely protects neurons from growth suppression under nutrient stress.

## FoxO is differentially expressed in C4da neurons and epidermal cells to regulate their distinct responses to nutrient stress

To further understand the mechanisms responsible for the differential growth regulations of epidermal cells and C4da neurons by nutrients, we first examined three genes known to inhibit mTor signaling under stress conditions, including *cryptocephal* (*crc*)/*ATF4* (*Kang et al., 2017*), *Sestrin* (*Sesn*) (*Lee et al., 2016*), and *Sirtuin 2* (*Sirt2*) (personal communications with Hening Lin). Anticipating that inhibiting the responsible genes in epidermal cells would relieve growth suppression under nutrient restriction, we knocked down the candidate genes in epidermal cells using *Gal4^{R16D01}*. However, knockdown of none of these genes in epidermal cells yielded a statistically significant increase of the cell size under either HY or LY condition (*Figure 4—figure supplement 1A*), indicating either that these genes do not suppress epidermal cell growth or that the RNAi lines were not effective. We also examined dendrite growth in a *Sirt2* null mutant and noticed nutrient-independent increases of normalized dendrite length compared to the control (*Figure 4—figure supplement 1B*). We next examined neuronal roles of two other components in the Tor pathway: Ras homolog enriched in brain (Rheb), a GTPase involved in Tor kinase activation (*Inoki et al., 2003*; *Saucedo et al., 2003*; *Tee et al., 2003*; *Zhang et al., 2003*), and Slimfast (Slif), an amino acid transporter that maintains the cellular amino acid level necessary for Tor activation (*Colombani et al., 2003*). Interestingly, knocking down *Rheb* in C4da neurons using a validated RNAi line (*Francis and Ghabrial, 2015*) caused a mild (11%) increase of dendrites in HY condition and a weak (11%) reduction in LY condition (*Figure 4—figure supplement 1C, D, F, G and I*), suggesting that Rheb suppresses dendrite growth under nutrient abundance but promotes dendrite growth under nutrient restriction. On the other hand, *slif* knockdown caused severe dendrite reduction in both HY (40%) and LY (74%) conditions (*Figure 4—figure supplement 1E, H and I*), suggesting that *slif* is required for proper dendrite growth regardless of the nutrient state. The fact that dendrite reduction in LY food is greater suggests that dendrite growth under nutrient restriction may rely more on the availability of amino acid transporters.

We further asked why epidermal cells exhibit more pronounced growth suppression than C4da neurons under nutrient stress by investigating the role of FoxO, because FoxO is an important cellular stress sensor that can impact mTor signaling (*Eijkelenboom and Burgering, 2013*). We first examined FoxO expression in neurons and epidermal cells using a *foxo-GFP* transgenic line that carries a 77 kb genomic fragment containing the full *foxo* locus with a C-terminal GFP tag and therefore should mimic the endogenous FoxO expression. In epidermal cells, FoxO-GFP showed cytoplasmic distribution under normoxia but translocated to epidermal nuclei within several minutes of hypoxia (*Figure 4—figure supplement 2A and B*), consistent with its known role as a transcription factor responsive to cellular stress (*Eijkelenboom and Burgering, 2013*). FoxO-GFP was expressed at similar levels in epidermal cells in both HY and LY conditions (*Figure 4A–C*) but could not be detected above the background noise level in C4da cell bodies (*Figure 4A–C*). To confirm these FoxO expression patterns, we stained larval body walls using a validated anti-FoxO antibody (*Slaidina et al., 2009*), which also showed much higher signals in epidermal cells than in C4da neurons (*Figure 4D* and *Figure 4—figure supplement 2C and D*). FoxO staining signals overlapped with those of FoxO-

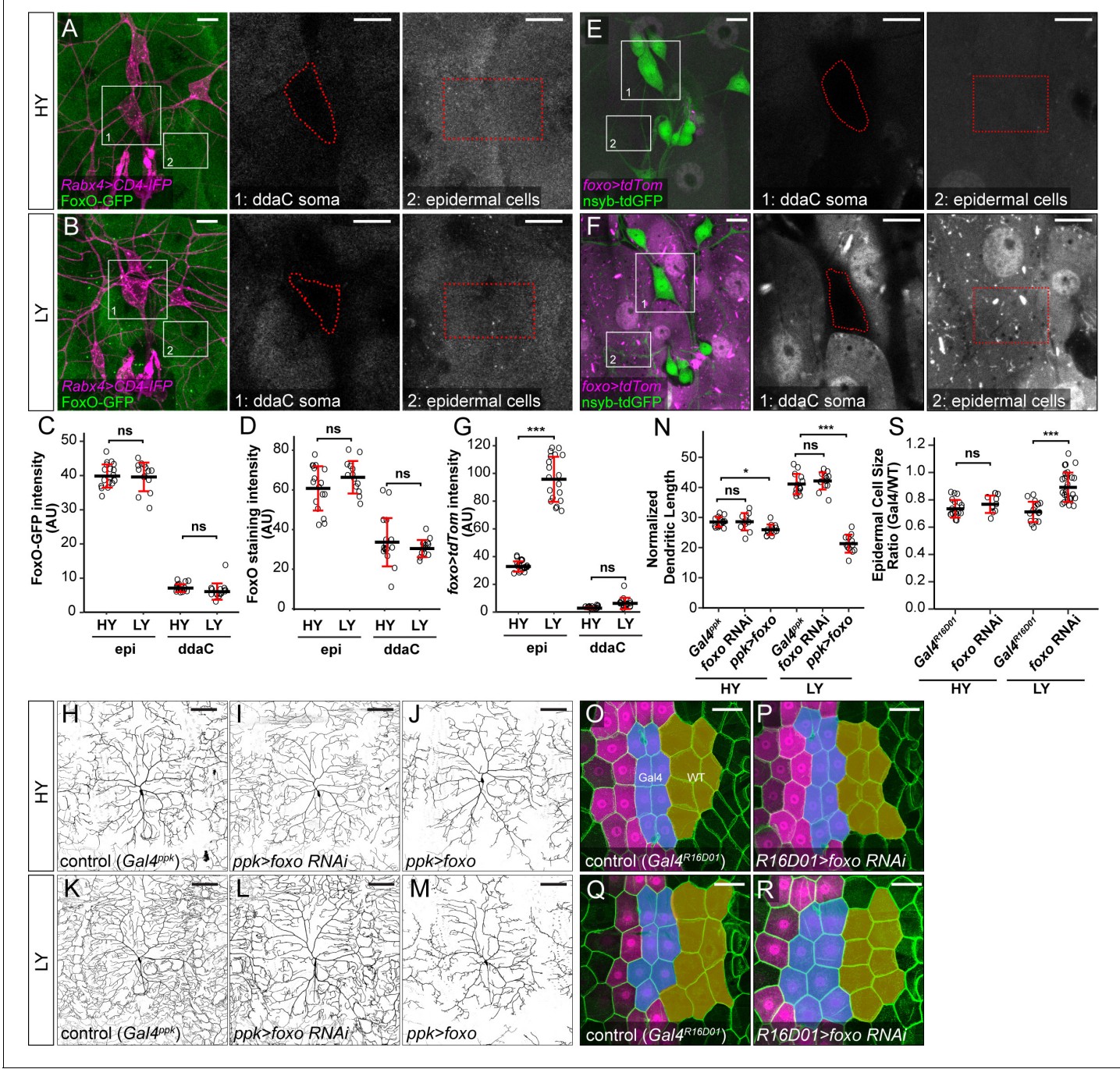

**Figure 4.** FoxO is differentially expressed in C4da neurons and epidermal cells to regulate their distinct responses to nutrient stress. (**A and B**) FoxO-GFP (green) in da neurons (magenta) and epidermal cells in HY and LY conditions in 2D projections. The insets for (**A**) and (**B**) show FoxO-GFP levels at ddaC somas (1) and epidermal cells (2) in single confocal sections. (**C**) Quantification of FoxO-GFP intensity in ddaC neuron soma and epidermal cells in HY and LY conditions. Two-way ANOVA. The differences between HY and LY in epidermal cells and ddaC are not significantly different as indicated by a non-significant interaction term (p=0.0901). Each circle represents a segment; epi: n = 18 for HY, n = 14 for LY; ddaC: n = 17 for HY and n = 14 for LY. (**D**) Quantification of FoxO staining in ddaC neuron soma and epidermal cells in HY and LY conditions. Each circle represents a segment; epi: n = 17 for HY, n = 13 for LY; ddaC: n = 17 for HY and n = 13 for LY. Two-way ANOVA. The differences between HY and LY in epidermal cells and ddaC are not significantly different as indicated by a non-significant interaction term (p=0.0897). (**E and F**) Gal4$^{foxo}$-driven tdTom (magenta) in da neurons (green) and epidermal cells in HY and LY conditions in 2D projections. The insets in (**E**) and (**F**) show Gal4$^{foxo}$-driven tdTom expression levels at ddaC somas (1) and epidermal cells (2) in single confocal sections. (**G**) Quantification of Gal4$^{foxo}$-driven tdTom intensity in ddaC neuron soma and epidermal cells in HY and LY conditions. Each circle represents a segment; epi: n = 19 for HY, n = 19 for LY; ddaC: n = 18 for HY, n = 18 for LY. (**H–M**) ddaC neurons in the Gal4$^{ppk}$ control (**H and K**) and animals expressing Gal4$^{ppk}$-driven foxo RNAi (**I and L**) and foxo (**J and M**) in HY and LY conditions. (**H**) and (**K**) are the

*Figure 4 continued on next page*

*Figure 4 continued*

same as *Figure 2A and D*, respectively. (**N**) Quantification of normalized dendritic length in control, *foxo* RNAi and *foxo* overexpression animals in HY and LY conditions. Two-way ANOVA, Posthoc contrasts with a Dunnett correction. The differences between control and *foxo* RNAi under HY and LY are not significantly different as indicated by a non-significant interaction term (p=0.5090), but it is significant between control and *ppk >foxo* as indicated by a significant interaction term (p=2e-16). Each circle represents a neuron; HY: n = 14 for *Gal4ppk*, n = 11 for *foxo* RNAi, n = 12 for *foxo* OE; LY: n = 14 for *Gal4ppk*, n = 13 for *foxo* RNAi, n = 12 for *foxo* OE. *Gal4ppk* is the same dataset as in *Figure 2G*. (**O–R**) Epidermal cells in the *Gal4R16D01* control and animals expressing *Gal4R16D01*-driven *Atg8a* RNAi in HY and LY conditions. (**O**) and (**Q**) are the same as *Figure 3D and F*, respectively. (**S**) Quantification of epidermal cell size ratio (*Gal4R16D01* cells/WT cells) in HY and LY conditions. Two-way ANOVA. The differences between control and *foxo* RNAi under HY and LY are significantly different as indicated by a significant interaction term (p=0.00023), Each circle represents a segment; HY: n = 18 for control, n = 10 for *foxo* RNAi; LY: n = 16 for control, n = 18 for *foxo* RNAi. For all quantifications, ***p<0.001; **p<0.01; *p<0.05; ns, not significant. Black bars, mean; red bars, SD. Scale bars, 10 µm in (A), (B), (E) and (F); 100 µm in (H–M); 50 µm in (O–R).

The online version of this article includes the following source data and figure supplement(s) for figure 4:

**Source data 1.** *foxo* analyses data for *Figure 4* and *Figure 4—figure supplements 1*, *2*, *3*.
**Figure supplement 1.** The effects of LOF of *Sesn, crc, Sirt2, Rheb,* and *slif* on C4da dendrite growth.
**Figure supplement 2.** FoxO expression.
**Figure supplement 3.** *foxO-Gal4* and effectiveness of *foxo* RNAi.

GFP in epidermal cells (*Figure 4—figure supplement 2E*) and were markedly reduced upon *foxo* knockdown (*Figure 4—figure supplement 2F*). Suspecting that the lack of nuclear FoxO-GFP in epidermal cells and neurons in LY media may be due to insufficient cellular stress, we subjected larvae to starvation or knocked down *slif* in all neurons to create nutrient stress. Within 4 hr of starvation, FoxO-GFP was drastically enriched in epidermal nuclei (*Figure 4—figure supplement 2H and J*). In the same experiment, FoxO-GFP remained undetectable in da neurons of 75% of larvae (*Figure 4—figure supplement 2H and J*) but increased to moderate levels in nuclei of da neurons of the rest of larvae (*Figure 4—figure supplement 2I and J*). In comparison, pan-neural *slif* knockdown severely impaired larval locomotion and caused nuclear accumulation of FoxO-GFP in epidermal cells (*Figure 4—figure supplement 2K–2M*). However, FoxO-GFP still could not be detected in nuclei of da neurons in these animals (*Figure 4—figure supplement 2K–2M*). These results together suggest that FoxO is differentially expressed and reacts differently to cellular stress in epidermal cells and C4da neurons.

To further investigate FoxO expression patterns, we converted a MiMIC insertion line of *foxo* into a *foxo-Gal4* using the Trojan exon technique (*Diao et al., 2015*), so that the Gal4 is under the same transcription regulation as the endogenous *foxo* (*Figure 4—figure supplement 3A*). This *foxo-Gal4* drove uniform epidermal expression of a *UAS-tdTom* reporter under HY, and the expression was enhanced 2.9 folds by nutrient stress (*Figure 4E–G*). In contrast, *foxo-Gal4* showed minimal activity in C4da neurons under both HY and LY conditions (*Figure 4E–G*), confirming the results obtained by FoxO-GFP and FoxO staining. Interestingly, *foxo-Gal4* is expressed in a subset of neurons and non-neural cells in the larval brain and ventral nerve cord (*Figure 4—figure supplement 3B and C*), suggesting cell-type-specific expression in the CNS as well.

Prior loss-of-function (LOF) and gain-of-function (GOF) studies of *foxo* in C4da neurons showed that FoxO plays a role in enhancing dendritic space-filling by stabilizing dendritic microtubule (*Sears and Broihier, 2016*). However, whether FoxO contributes to nutrient regulation of dendrite growth is unclear. Because *foxoΔ94*, a null mutation of *foxo* (*Slack et al., 2011*), caused larval lethality before the 3rd instar in both HY and LY media, we chose to knock down or overexpress *foxo* in C4da neurons. *foxo* knockdown in epidermal cells effectively eliminated *UAS-FoxO-GFP* expression (*Figure 4—figure supplement 3D–3F*), confirming the efficacy of this RNAi line. We found that knocking down *foxo* in C4da neurons had no effect on normalized dendrite length in either condition (*Figures 4H, I, K, L and N*), but caused 13% reduction of dendrite density in LY medium. This decrease in dendrite density was caused in part by an expansion of dendritic fields of *ppk >foxo* RNAi neurons, which occupied a larger 2-dimensional area under LY conditions, despite exhibiting total dendrite lengths comparable to controls. On the other hand, overexpressing FoxO in C4da neurons caused mild (11.7%) and strong (49.2%) reduction of normalized dendrite length in HY and LY conditions, respectively (*Figures 4J, M and N*). Lastly, we examined the role of FoxO in epidermal cell growth by knocking down *foxo* using *Gal4R16D01*. The knockdown had no effect in HY but increased the epidermal cell size by 34% under the LY condition (*Figure 4O–S*).

The above results collectively suggest that FoxO differentially affects the growth of C4da neurons and epidermal cells. In neurons, although high FoxO levels are growth-inhibitory, endogenous FoxO is likely expressed at levels too low to directly affect dendrite growth; in epidermal cells, FoxO is more highly expressed, but inhibits cell growth only under nutrient stress. Our results also suggest that among genes known to inhibit mTor signaling under nutrient stress, *foxo* plays a more significant role in nutrient-dependent dendrite overgrowth of somatosensory neurons. We therefore chose to focus our further analysis on how FoxO overexpression suppresses dendrite growth.

## Overexpressed FoxO exerts its effects through modulating Tor signaling and autophagy

Because the relative strength of Tor signaling in C4da neurons as compared to epidermal cells determines the extent of dendrite innervation in the epidermis (*Figure 2*), we used pRpS6 staining to examine whether overexpressed FoxO affects dendrite growth through suppressing Tor signaling. In HY, neuronal FoxO overexpression did not cause detectable changes in the ratio of pRpS6 levels between neuronal compartments and epidermal cells (*Figures 2M*, *5A, C and D*). However, in LY, while pRpS6 levels in the somas of FoxO-overexpressing neurons were still higher than in epidermal cells, the dendritic pRpS6 signal was reduced to levels lower than those of epidermal cells (*Figure 5B–D*). These data suggest that high levels of FoxO specifically suppress Tor signaling in dendrites in a nutrient-dependent manner. We further examined the autophagy level of FoxO-overexpressing neurons using mCherry-Atg8a, as higher autophagy levels suppress the growth of both C4da neurons and epidermal cells (*Figure 3*). While the autophagosome level in C4da cell bodies was not altered by FoxO-expression in HY, it increased 4.9 folds in LY (*Figure 4E–G*). The above results together suggest that nutrient stress enables overexpressed FoxO to suppress dendritic Tor signaling and to induce autophagy in neurons. The lack of high FoxO expression in wildtype neurons thus ensures preferential dendrite growth in nutrient stress by protecting dendritic Tor signaling and suppressing autophagy.

## Nutrient stress-induced dendrite overgrowth sensitizes neurons

C4da neurons are polymodal nociceptive neurons that respond to noxious thermal, mechanical, chemical, and light stimuli (*Tracey et al., 2003*; *Hwang et al., 2007*; *Xiang et al., 2010*). To test whether the preferential dendrite growth under nutrient stress is physiologically relevant, we examined whether nutrient stress influences larval nociception using an established heat-response assay (*Babcock et al., 2009*). In this assay, a temperature-controlled heat probe is used to deliver noxious thermal stimuli that elicit C4da-dependent larval rolling behavior. We first examined wildtype larvae reared in HY and LY media. Over a range of temperatures (from 40°C to 46°C), we found that a larger percentage of LY larvae exhibited nocifensive rolling responses to thermal stimuli than HY larvae (*Figure 6A*). The temperatures required to induce response in a similar percentage of larvae were approximately 2°C lower for the LY condition than for the HY condition. At temperatures above 46°C the vast majority of larvae in both conditions exhibited nocifensive rolling. We additionally monitored response latency and found that significantly more larvae in LY responded within 5 s (fast response) of heat stimulus over a temperature range from 43°C to 48°C (*Figure 6B*). When we examined response latency at 46°C with higher temporal resolution, we found that the majority of LY larvae responded within 6 s while HY responses were distributed over a broader range of durations (*Figure 6C*). These data suggest that nutrient stress sensitizes larvae so that they react more acutely to noxious heat.

We next examined the effects of FoxO overexpression in C4da neurons by stimulating the larvae at 46°C. While FoxO overexpression did not significantly change the rolling behaviors of HY larvae, it significantly reduced the percentages of LY larvae that showed fast responses and overall response (*Figure 6D and E*). This decrease of nociceptive response thus correlates with the strong dendrite reduction of FoxO-overexpressing neurons under nutrient restriction, even though we cannot rule out the possibility that FoxO overexpression impairs neuronal function through other means.

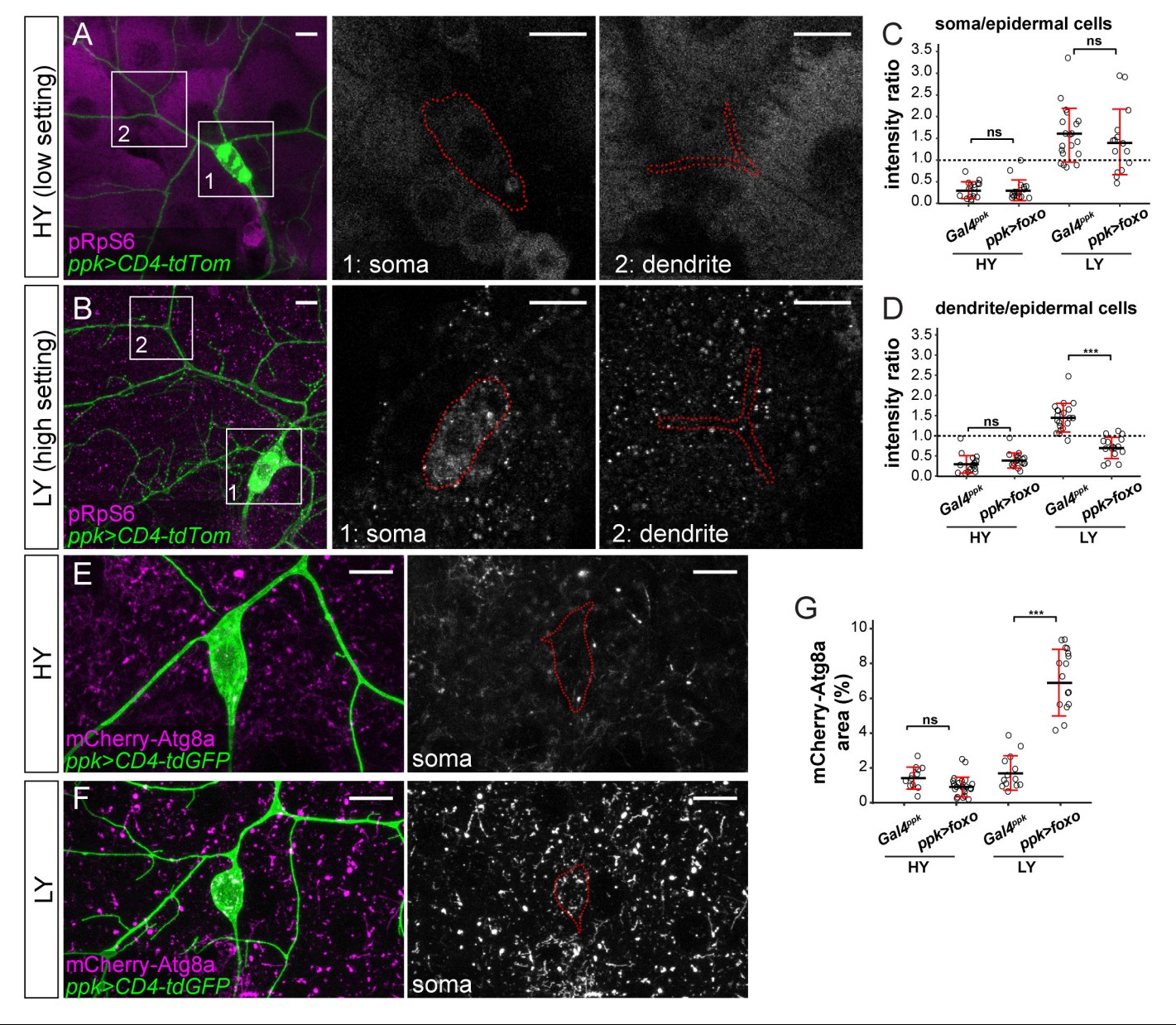

**Figure 5.** FoxO exerts its effects through modulating Tor signaling and autophagy. (**A and B**) pRpS6 staining (magenta) of ddaC neurons (green) and epidermal cells in *Gal4$^{ppk}$*-driven *UAS-foxo* animals in HY and LY conditions in 2D projections. The insets for (**A**) and (**B**) show pRpS6 staining at the soma (1) and primary dendrites (2) in single confocal sections, with the somas and dendrites outlined. (**C and D**) Quantification of pRpS6 intensity ratios in control and *foxo* OE animals in HY and LY conditions. For both, Two-way ANOVA. The differences between control and *ppk >foxo* are not significant between the HY and LY in soma, as indicated by a non-significant interaction term (p=0.9584), but are significant for dendrite, as indicated by a significant interaction term (p=9.66e-09). Each circle represents a segment; HY: n = 17 for control, n = 17 for *foxo* RNAi; LY: n = 20 for control, n = 16 for *foxo* RNAi. The control datasets are the same as in *Figure 2P*. (**E and F**) mCherry-Atg8a (magenta) in ddaC soma (green) of *Gal4$^{ppk}$*-driven *UAS-foxo* animals in HY and LY. The soma images are projections from thinner volumes only containing the soma. (**G**) Quantification of mCherry-Atg8a levels in ddaC somas measured by the area percentage of mCherry-Atg8a-positive vesicles. Two-way ANOVA. The differences between control and *ppk >foxo* are significantly different between the HY and LY, as indicated by a significant interaction term (p=5.51e-09). Each circle represents a neuron; HY: n = 12 for control, n = 23 for *foxo* OE; LY: n = 13 for control, n = 19 for *foxo* OE. For all quantifications, ***p<0.001; ns, not significant. Black bars, mean; red bars, SD. Scale bars, 10 μm.

The online version of this article includes the following source data for figure 5:

**Source data 1.** FoxO OE data for *Figure 5*.

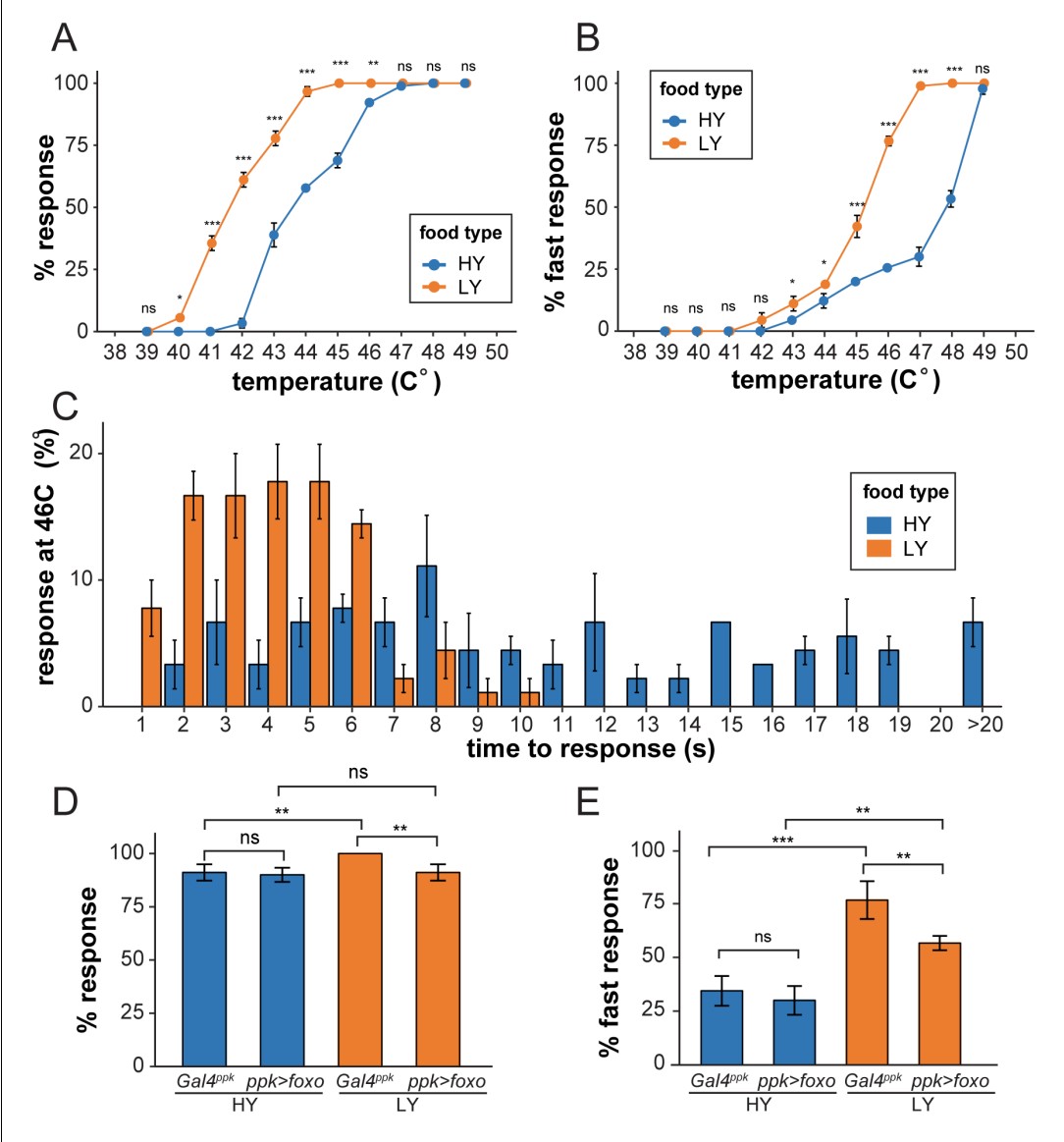

**Figure 6.** Nutrient stress-induced dendrite overgrowth sensitizes neurons. (**A**) A plot showing the percent of responders (respond within 20 s) versus temperature. n = number of larvae; n = 90 for HY and LY at each temperature. (**B**) A plot showing the percent of fast responders (respond within 5 s) versus temperature. n = number of larvae; n = 90 for HY and LY at each temperature. (**C**) A plot showing the percent of responders at 46°C versus the response time. n = number of larvae; n = 90 for HY and LY. (**D**) A plot showing the percent of responders versus temperature for *Gal4^ppk* and FoxO OE animals in HY and LY conditions at 46°C. n = number of larvae; n = 120 for each group. Two-way ANOVA; interaction term between genotype and nutrient, p=0.0678. (**E**) A plot showing the percent of fast responders versus temperature for *Gal4^ppk* and FoxO OE animals in HY and LY conditions at 46°C. n = number of larvae; n = 120 for each group. Two-way ANOVA; interaction term between genotype and nutrient, p=0.08052. In all panels, error bars indicate the standard error from three repeats (SE). For all quantifications, ***p<0.001; **p<0.01; *p<0.05: ns, not significant; Two-way ANOVA. The online version of this article includes the following source data for figure 6:

**Source data 1.** Behavior analyses data for *Figure 6*.

## Different types of somatosensory neurons respond to nutrient stress differentially

Among the four classes of da neurons, C4da neurons have dendritic arbors that exhibit space-filling and are highly dynamic (*Grueber et al., 2003*; *Poe et al., 2017*), while C1da and C2da neurons have simple arbors and sparse dendrites occupying defined territories and grow mostly by scaled expansion of the existing dendritic arbors established in late embryogenesis (*Grueber et al., 2002*). C3da

neurons grow more complex dendritic arbors and are characterized by numerous short terminal branches called dendritic spikes, which are highly dynamics during larval growth (*Grueber et al., 2002*; *Nagel et al., 2012*). We asked whether nutrient stress also impacts dendritic growth of C1da and C3da neurons. Consistent with the previous report by *Watanabe et al., 2017*, we found that the yeast concentration did not have obvious effects on the total dendrite length of C1da neuron ddaE (*Figure 7A–C*). However, nutrient stress stimulated the dendritic growth of C3da neurons ddaA and ddaF: the total dendrite length increased by 40% and 32%, respectively; the total terminal dendrite length increased by 61% and 46%, respectively; the terminal branch numbers increased by 47% and 58%, respectively (*Figure 7D–H*). These data suggest that nutrient stress promotes over-branching of complex dendritic arbors of C3da and C4da neurons but not simple arbors of C1da neurons.

We then examined whether C1da and C3da neurons are subjected to the same regulation of autophagy as C4da neurons under nutrient stress. Interestingly, the cell bodies of these neurons showed variable and cell-specific baseline autophagy levels in HY, as indicated by mCherry-Atg8a, but these levels did not appear to be altered by the nutrient level (*Figure 7I–M*). Similarly, C3da neurons showed low and nutrient-independent lysosomal level (*Figures 7O, Q and R*). Interestingly, C1 ddaE neurons showed a much higher level of the lysosomal marker Lamp-mCherry than other da classes in HY, and this level was enhanced 2.7 folds by nutrient stress (*Figures 7N, P and R*), suggesting that C1da neurons have uniquely high and nutrient-dependent lysosomal system. Lastly, we examined *foxo* expression in C1da and C3da neurons using *foxo-GFP* and *foxo-Gal4*. Similar to C4da neurons, C1da and C3da neurons showed only background-noise levels of FoxO-GFP and *foxo >tdTom* signals (*Figure 7M and N*). These data suggest that, similar to C4da neurons, C1da and C3da neurons exhibit low FoxO expression and nutrient-independent autophagy levels, while C3da neurons, but not C1da neurons, show nutrient stress-induced dendrite overgrowth.

## Discussion

In this study, we show that *Drosophila* C3da and C4da neurons exhibit a growth advantage over neighboring epidermal cells under nutrient restriction, resulting in dendrite overgrowth. This tissue-specific growth regulation by nutrient stress is at least partially determined by FoxO expression level (*Figure 8*). In non-neural tissues like epidermal cells, the stress sensor FoxO is expressed at sufficient levels to allow epidermal cells to respond robustly to nutrient stress. In these tissues, high nutrition elevates InR/Tor signaling and suppresses FoxO activity and autophagy, leading to a high growth rate; in low nutrients, the reduction in InR/Tor signaling combined with high FoxO activity stimulates autophagy and greatly slows down cell size increase. In contrast, PNS neurons express FoxO at much lower levels and exhibit low basal levels of autophagy. As a result, dendritic InR/Tor signaling is not further suppressed by FoxO when the systemic insulin level is low. Therefore, the low FoxO expression and the lack of autophagy induction in PNS neurons protect dendrite growth from nutrient stress. Interestingly, Rheb in neurons mildly suppresses dendrite growth in high nutrition but enhances dendrite growth under nutrient stress. Whether these effects are mediated by Tor remains to be determined.

Brain sparing has been recognized in both mammals and insects as an important means to protect the developing nervous system from nutrient deficiency. The preferential dendrite growth of da neurons constitutes another form of nervous system sparing that bears important distinctions from *Drosophila* brain sparing. First, while proliferating neural stem cells in the CNS is protected against starvation (*Cheng et al., 2011*), individual post-mitotic da neurons are spared at the level of neuronal arbor growth. Second, unlike CNS neuroblasts, which rely on a special extrinsic factor (the glial-derived Jeb) to sustain neurogenesis, PNS neurons possess a unique intrinsic genetic program that endows them the resistance to nutrient stress. Lastly, the CNS sparing is independent of InR and Tor, made possible by the alternative Jeb/Alk/PI3K pathway, while the PNS protection still relies on InR/Tor signaling. Therefore, our work reveals a novel mechanism of neural protection under nutrient stress.

In the mammalian brain, FoxO members are highly expressed in NSCs and are required for the long-term maintenance of the adult NSC pool critical for adult neurogenesis (*Paik et al., 2009*; *Renault et al., 2009*; *Yeo et al., 2013*). Recently, FoxOs were also found to regulate dendrite branching and spine density of adult-born hippocampal neurons (*Schäffner et al., 2018*).

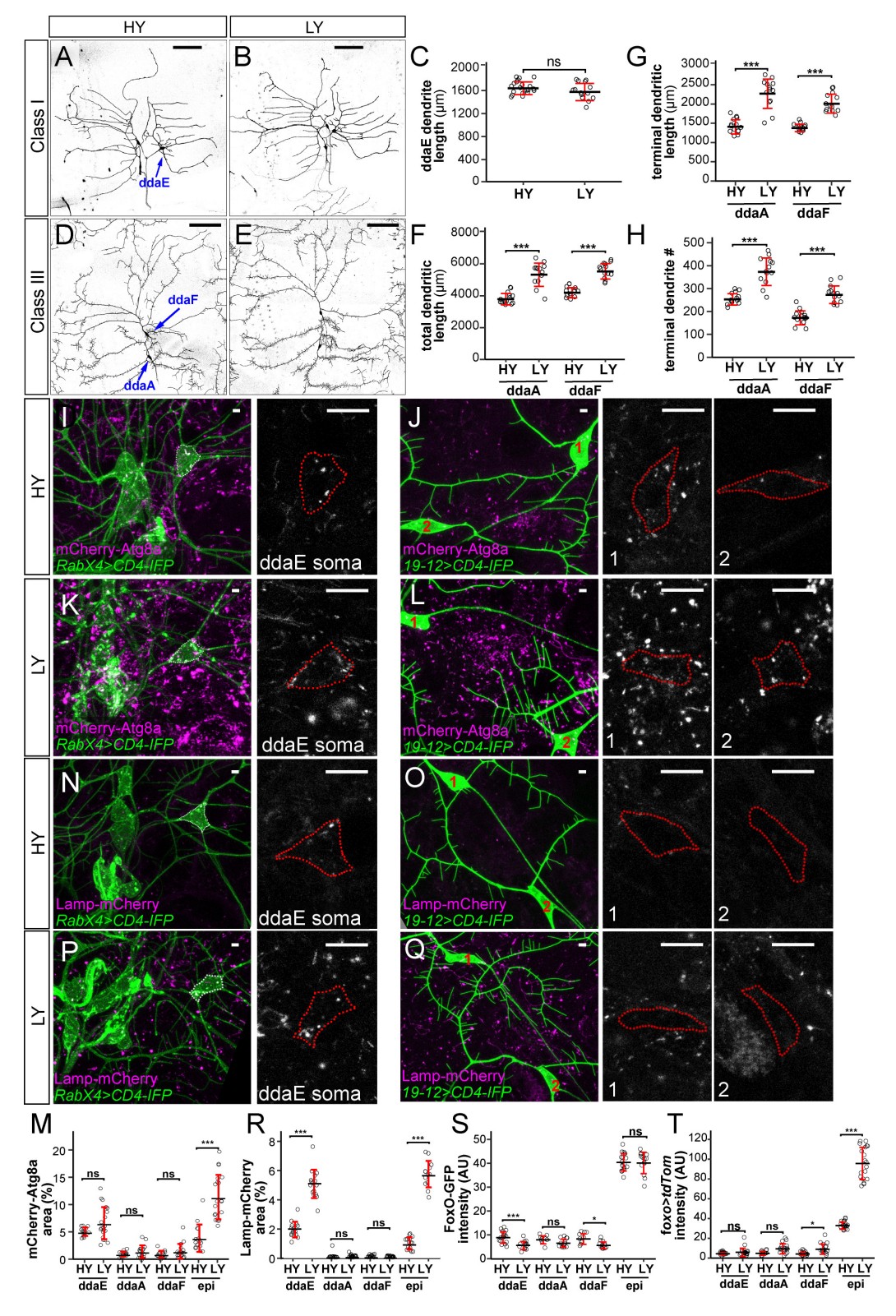

**Figure 7.** Different types of somatosensory neurons respond to nutrient stress differentially. (**A and B**) C1da neurons in HY and LY conditions. (**C**) Quantification of ddaE dendrite length in HY and LY conditions. Each circle represents a neuron; n = 20 for HY, n = 14 for LY. student's t-test. (**D and E**) C3da neurons in HY and LY conditions. (**F–H**) Quantification of total dendritic length (**F**), terminal dendritic length (**G**) and terminal dendrite number (**H**) of ddaF and ddaA neurons in HY and LY conditions. Each circle represents a neuron; ddaA: n = 14 for HY, n = 13 for LY; ddaF: n = 14 for HY, n = 13 for

*Figure 7 continued on next page*

*Figure 7 continued*

LY. (I–L) mCherry-Atg8a (magenta) in da neuron somas (green) in HY and LY conditions. (I) and (K) show ddaE. (J) and (L) show ddaF (1) and ddaA (2). The soma images are projections from thinner volumes only containing the soma. The somas are outlined. (M) Quantification of mCherry-Atg8a levels in da neuron somas and epidermal cells in HY and LY conditions, measured by the area percentage of mCherry-Atg8a-positive vesicles. Each circle represents a segment; HY: n = 15 for ddaE, n = 15 for ddaA, n = 15 for ddaF, n = 14 for epi; LY: n = 20 for ddaE, n = 18 for ddaA, n = 18 for ddaF, n = 20 for epi. (N–Q) Lamp-mCherry (magenta) in da neuron somas (green) in HY and LY conditions. (N) and (P) show ddaE. (O) and (Q) show ddaF (1) and ddaA (2). The soma images are projections from thinner volumes only containing the soma. The somas are outlined. (R) Quantification of Lamp-mCherry levels in da neuron somas and epidermal cells in HY and LY conditions, measured by the area percentage of Lamp-mCherry-positive vesicles. Each circle represents a segment; HY: n = 15 for ddaE, n = 12 for ddaA, n = 15 for ddaF, n = 15 for epi; LY: n = 16 for ddaE, n = 16 for ddaA, n = 16 for ddaF, n = 16 for epi. (S) Quantification of FoxO expression level in da neuron soma in HY and LY conditions. Each circle represents a segment; LY: n = 14 for ddaE, n = 13 for ddaA, n = 13 for ddaF, n = 14 for epi; HY: n = 17 for ddaE, n = 9 for ddaA, n = 11 for ddaF, n = 18 for epi. (T) Quantification of *Gal4^foxo*-driven *tdTom* expression levels in da neuron somas in HY and LY conditions. Each circle represents a segment; HY: n = 19 for ddaE, n = 13 for ddaA, n = 18 for ddaF, n = 19 for epi; LY: n = 17 for ddaE, n = 16 for ddaA, n = 17 for ddaF, n = 17 for epi. For all quantifications, ***p<0.001; **p<0.01; *p<0.05; ns, not significant. Student's t-test. Black bars, mean; red bars, SD. Scale bars, 100 µm in (A), (B), (D) and (E); 10 µm in (I–L) and (N–Q). Epi datasets in (S) and (T) are the same as the ones in *Figure 4C and G*, respectively.

The online version of this article includes the following source data for figure 7:

**Source data 1.** Other da neuron classes data for *Figure 7*.

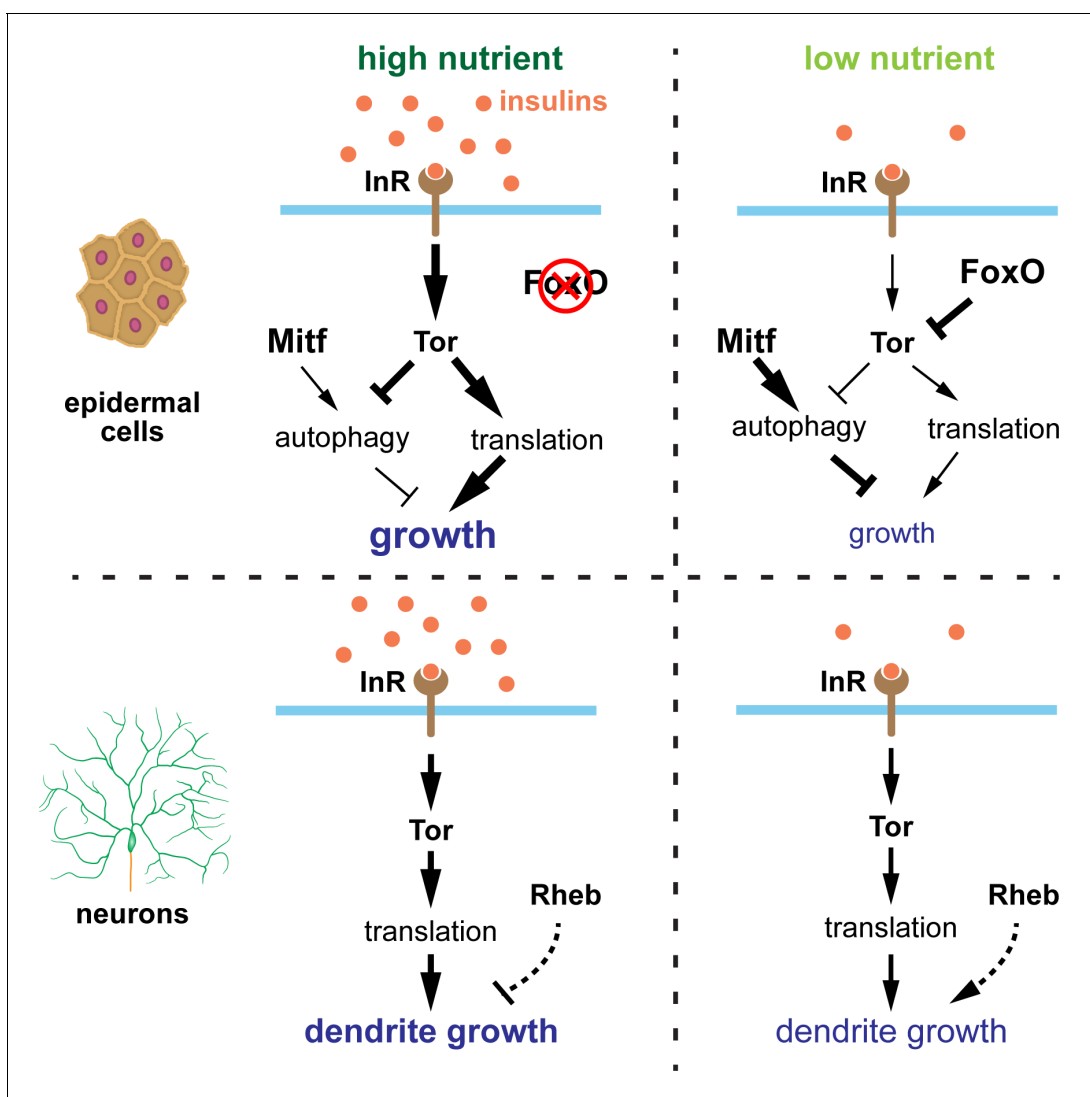

**Figure 8.** Model of nutrient-regulated da neuron and epidermal cell growth. See discussion for details. Supplementary file - Key Resources Table.

Interestingly, FoxOs in these neurons suppress mTor signaling and maintain a level of autophagic flux that is necessary for the normal morphogenesis of the neurons. In addition, FoxOs were found to be upregulated in aged brains and function to delay aging-related axonal tract degeneration by suppressing mTor activity (*Hwang et al., 2018*). In *Drosophila* C4da neurons, FoxO was previously found to promote dendrite space-filling and to mediate polyQ-induced neuronal toxicity (*Sears and Broihier, 2016*; *Kwon et al., 2018*).

While these prior studies established key roles for FoxO proteins in nervous system development, the relationship between nutrition and neuronal FoxO function was previously unexplored. Likewise, FoxO expression patterns in neural and non-neural tissues have not been compared. Our results demonstrate that FoxO is expressed at a much lower level in da neurons than in non-neural larval tissues. Consequently, inhibiting *foxo* in neurons does not directly affect dendrite growth, even though overexpressing FoxO in neurons inhibits dendrite growth. Nevertheless, our results support that neuronal FoxO mildly promotes dendritic space-filling of C4da neurons only under nutrient restriction by reducing the size of the dendrite field though an unknown mechanism. More importantly, the lack of high FoxO expression makes neurons insensitive to nutrient stress, giving them a growth advantage over non-neural tissues that express higher levels of FoxO. Interestingly, this FoxO-dosage-dependent nutrient-insensitivity has been previously described in the adult genitalia (*Tang et al., 2011*). While adult wings and maxillary palps become small on poor diets, the size of genital arches is affected much less; these differential responses are linked to tissue-intrinsic levels of FoxO expression. However, in this example and previously described FoxO functions in growth regulation, FoxO primarily regulates cell numbers but not cell size (*Jünger et al., 2003*; *Puig et al., 2003*). Therefore, our study reveals a novel function for FoxO in environmental regulation of neural development. It is worth noting that although FoxO is minimally expressed in da neurons, it is expressed at a higher level in a subpopulation of CNS neurons, raising the possibility that the arbor growth of these neurons may be differentially regulated by nutrient availability.

Lastly, our results reveal a level of neuronal diversity in the response to nutrient stress. Although all da neurons we examined show similarly low FoxO expression and the lack of autophagy induction under nutrient restriction, only class III and IV but not class I neurons display preferential dendrite growth. This distinction may be related to their arbor growth mechanisms and neuronal functions. As proprioceptive neurons that detect body surface folds during locomotion (*He et al., 2019*; *Vaadia et al., 2019*), C1da neurons have simple arbors covering defined territories on the larval body wall. Their dendritic arbors grow mainly by expanding the shafts of the dendritic branches established during embryogenesis. Their functional demands may require a tighter growth coupling with the epidermis but not with environmental nutrition availability. In contrast, C3da and C4da neurons have highly dynamic high-order branches that grow also by branching and tip extension. The lack of growth suppression therefore leads to dendrite hyperarborization. As these neurons sense mild to extreme levels of external stimuli, the heightened sensations allowed by their dendrite overgrowth may bestow the larva a survival advantage in an unfavorable environment.

## Materials and methods

### Live imaging

Animals were raised at 25℃ in density-controlled vials containing between 50 and 70 embryos collected in a 3 hr time window. To achieve optimum embryo densities, approximately 50 virgin females were aged 5 days on molasses food with yeast paste, crossed with approximately 15–20 males, and then allowed to mate for 1–2 days on molasses food with yeast paste. Embryo collections were then performed in a 3 hr time window on both 1% and 8% yeast food. Third instar larvae at 86 hr AEL on 8% yeast or 216 hr AEL on 1% yeast were mounted in glycerol and imaged with a Leica SP8 confocal. The A2-A3 segments of 8–10 larvae were imaged for each genotype using a 20X oil objective. To image larvae younger than 72 hr AEL, larvae were anesthetized by isoflurane for 2 min and then mounted in halocarbon oil. To image FoxO-GFP under normoxia, larvae were handled with care and imaged within 2 min after mounting on slides. For imaging under hypoxia, larvae were left on the slides for 5 min before imaging.

## Fly food recipe

Fly food was prepared using the following recipes (for the dispersal of ~12 mL into 20 vials).

| Ingredients: | 1% yeast (LY) | 8% yeast (HY) |
| --- | --- | --- |
| Distilled $H_2O$ | 240 mL | 234 mL |
| Agar | 2.4 g (12 g/L) | 2 g (10 g/L) |
| Glucose | 20 g | 20 g |
| Inactive yeast | 2.5 g | 20 g |
| Acid mix (phosphoric acid + propionic acid) | 2 mL | 2 mL |
| Target final solution volume | 250 mL | 250 mL |

Acid Mix was made by preparing Solution A (41.5 ml Phosphoric Acid mixed with 458.5 ml distilled water) and Solution B (418 ml Propionic Acid mixed with 82 ml distilled water) separately and mixing Solution A and Solution B together.

## Fly stocks

The strains used in this study are listed in the Key Resources Table. We used the following neuronal markers to label specific classes of da neurons: *ppk-CD4-tdGFP*, *ppk-Gal4*, *UAS-CD4-tdGFP* and *UAS-CD4-tdTom* for C4da (*Han et al., 2011*; *Han et al., 2012*); *R10D05-CD4-tdTom* for C1da; *NompC-LexA::p65 LexAop-CD4-tdTom* and *Gal4[19-12] UAS-CD4-tdGFP repo-Gal80* for C3da (*Awasaki et al., 2008*; *Rumpf et al., 2011*); *RabX4-Gal4 UAS- mIFP-2A-HO1* for all classes of da neurons. *ppk-Gal4* was used for RNAi knockdown and overexpression in C4da neurons. *RabX4-Gal4* was used for RNAi knockdown in all neurons. *Gal4[R38F11]* and *Gal4[R16D01]* were used for RNAi knockdown and overexpression in all or stripes of epidermal cells, respectively.

*foxo-Gal4[MI00786]* was generated using Trojan-MiMIC system (*Diao et al., 2015*). MiMIC line MI00786 was crossed to flies with the triplet Trojan donor construct. The progeny of this cross were then crossed to females expressing Cre recombinase and ΦC31 integrase in the germline which allow the Trojan exons to replace the MiMIC attP cassettes. Progeny were then crossed to *UAS-GFP* for screening Gal4 expression by fluorescence microscopy. Adults positive for GFP expression were used to establish the line. The 2A-Gal4 insertions from established lines were sequenced to confirm the accuracy of the site and the reading frame. *nsyb-tdGFP* (*Poe et al., 2019*) was used to visualize neuronal cell bodies in the larval brain when *foxo-Gal4* expression was examined.

## Immunostaining

Antibody staining was done as previously described (*Poe et al., 2017*). Briefly, third instar larvae were dissected in cold PBS, fixed in 4% formaldehyde/PBS for 20 min at room temperature, and stained with the proper primary antibodies and subsequent secondary antibodies, each for 2 hr at room temperature. The details of the antibodies used are in the Key Resources Table.

## Image quantification

### Neuron quantification

Unless noted otherwise, only larvae with the segment width falling between 500 and 550 µm were quantified for neurons. The tracing and measurement of da neuron dendrites were done in Fiji/ImageJ as previously described (*Poe et al., 2017*). Briefly, images of dendrites (1,024 × 1,024 pixels) taken with a 20X objective were first processed sequentially by Gaussian Blur, Auto Local Threshold, Particles4, Skeletonize (2D/3D), and Analyze Skeleton (2D/3D) plugins. The length of skeletons was calculated based on pixel distance. Dendrite density was calculated using the formula: 1000 X dendritic length (µm)/dendritic area (µm$^2$); normalized dendrite length was calculated as dendritic length (µm)/segment width (µm). Normalized length ratio was calculated using the formula: normalized dendrite length on LY/normalized dendrite length on HY.

## Epidermal cell quantification

Images of epidermal cells labeled by Nrg-GFP (1,024 × 1,024 pixels) taken with a 20X objective were first processed by Gaussian Blur (Sigma: 1) and then Auto Local Threshold (Phansalkar method, radius: 30). Isolated particles below the size of 500 pixels were removed by the Particles4 plugin (http://www.mecourse.com/landinig/software/software.html). The Nrg-GFP signal was then converted to single-pixel-width skeletons of epidermal cell borders using the Skeletonize (2D/3D) plugin. Images were then visually inspected to ensure that all epidermal cell borders were accurately labeled. Any erroneous epidermal cell borders were removed. Regions of interest (ROIs) were manually drawn to encompass the epidermal cells for quantification. Analyze Particles was then used to measure area, perimeter, height (Feret), and width (minFeret) for each epidermal cell in the ROIs. Epidermal cell size ratio was calculated as average cell size in the RNAi-expressing region/average cell size in the WT region.

## Other image quantification

For quantification of Lamp-mCherry and Atg8a-mCherry in Fiji/ImageJ, Z-stack images of dendrites and Lamp-mCherry or Atg8a-mCherry (1,024 × 1,024 pixels) taken with a 40X objective and a step size of 0.5 μm were converted into binary masks using thresholding. For quantification in epidermal cells, the stacks were projected into 2D images. ROIs were drawn manually outside the neuron. For quantification in neurons, care was taken to select the optical sections only containing the cell body but not Lamp-mCherry or Atg8a-mCherry signals below or above. The optical sections were then projected into 2D images, and ROIs covering cell bodies were generated based on masks of the dendrite channel. Finally, the mask area percentage within each ROI was calculated.

Images of *Mitf-GFPnls* (1,024 × 1,024 pixels) were taken with a 40X objective. A ROI on neuronal or epidermal cell nucleus was drawn manually on a single slice with the strongest signal. The mean gray value of the area was calculated.

Autophagic flux levels were measured by Atg8a-mCherry area percentage divided by GFP intensity. Images of epidermal cells and ddaC neurons (1,024 × 1,024 pixels) were taken with a 40X objective and a step size 0.5 μm. Atg8a-mCherry area percentages were determined as described above. For GFP intensity, a ROI in ddaC neuron cell body or epidermal cells were drawn manually using the maximum projected image and the mean gray value of the area was calculated.

## Statistics

For statistical comparison of two samples (e.g. WT vs. KD) and comparison of two conditions (HY vs. LY) in each tissue, Student's t-test was used. For results with one independent variable (e.g. genotype), one-way analysis of variance (ANOVA) with Tukey's HSD test was used. For results with two variables (e.g. genotype and nutrient condition), the data were analyzed using two way ANOVA with genotype, nutrient condition and their interaction term. Posthoc contrasts with a Dunnett correction for multiple comparisons were used. For results of interaction terms (genotype: nutrient condition), two-way ANOVA was used. For additional information on number of samples, see figure legends. R studio was used for all statistical analyses.

## Behavior assay

Larval heat-induced pain response was measured as described previously (*Babcock et al., 2009*). Wandering third larvae were scooped out of the food and gently cleaned with water, then transferred into a small petri dish with water drops to keep the animals moist. A temperature-controlled heat probe (ProDev Engineering, TX) was used to apply the heat onto the larval body surface. The stimulus was delivered by gently touching the animals laterally on segment A4. Each animal can only be tested once. The response latency was measured from the start of touch on the animal until it initiated a complete 360° roll.

## Acknowledgements

We thank Francesca Pignoni, Gábor Juhász, Heather Broihier, and Bloomington Stock Center for fly stocks; Jongkyeong Chung and Pierre Leopold for antibodies; Cornell CSCU for advice on statistics; Michael Goldberg and Jay Parrish for critical reading and suggestions on the manuscript. This work

was supported by a Cornell start-up fund and NIH grants (R01NS099125 and R21OD023824) awarded to CH.

## Additional information

### Funding

| Funder | Grant reference number | Author |
|---|---|---|
| National Institute of Neurological Disorders and Stroke | R01NS099125 | Chun Han |
| NIH Office of Research Infrastructure Programs | R21OD023824 | Chun Han |
| Cornell University | Start-up fund | Chun Han |

The funders had no role in study design, data collection and interpretation, or the decision to submit the work for publication.

### Author contributions

Amy R Poe, Conceptualization, Resources, Data curation, Formal analysis, Supervision, Validation, Investigation, Visualization, Methodology; Yineng Xu, Conceptualization, Data curation, Software, Formal analysis, Supervision, Validation, Investigation, Visualization, Methodology; Christine Zhang, Kailyn Li, Data curation, Formal analysis, Investigation, Visualization; Joyce Lei, Formal analysis, Investigation; David Labib, Investigation, Data acquisition and analysis; Chun Han, Conceptualization, Resources, Data curation, Software, Supervision, Funding acquisition, Project administration

### Author ORCIDs

Yineng Xu (iD) https://orcid.org/0000-0002-4473-4052
Chun Han (iD) https://orcid.org/0000-0001-7319-8095

### Decision letter and Author response

Decision letter https://doi.org/10.7554/eLife.53351.sa1
Author response https://doi.org/10.7554/eLife.53351.sa2

## Additional files

### Supplementary files

- Supplementary file 1. Key Resources Table.
- Transparent reporting form

### Data availability

All data generated or analysed during this study are included in the manuscript and supporting files.

The following previously published datasets were used:

| Author(s) | Year | Dataset title | Dataset URL | Database and Identifier |
|---|---|---|---|---|
| Rubin Lab | 2014 | Expression Patterns of GAL4 and LexA Driver Lines - GAL4: 26656, LexA: 23821 | https://flweb.janelia.org/cgi-bin/view_flew_imagery.cgi?line=R16D01 | Fly Light, R16D01 |
| Rubin Lab | 2014 | Expression Patterns of GAL4 and LexA Driver Lines - GAL4: 28802, LexA: 35137 | https://flweb.janelia.org/cgi-bin/view_flew_imagery.cgi?line=R38F11 | Fly Light, R38F11 |

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
