## [Decision Letter]

**Acceptance summary:**

This work describes important and novel findings related to dendritic sparing when larvae are starved and documents the role of FoxO as a key intrinsic transcriptional program in this process convincingly.

**Decision letter after peer review:**

Thank you for submitting your article "Low FoxO expression in *Drosophila* somatosensory neurons protects dendrite growth under nutrient restriction" for consideration by *eLife*. Your article has been reviewed by three peer reviewers, and the evaluation has been overseen by a Reviewing Editor and K VijayRaghavan as the Senior Editor, Hugo Bellen. The following individuals involved in review of your submission have agreed to reveal their identity: Fengwei Yu (Reviewer #1).

As you will notice, the reviews are quite positive and thorough. The reviewers have extensively discussed their reviews with one another and the Reviewing Editor has drafted this decision to help you prepare a revised submission.

We are enclosing the full report of the three reviewers as they are each useful and provide context. However, below we provide a summary of the key points that need to be addressed as agreed upon by the reviewers in their exchanges.

Essential revisions:

The authors make some comparisons between the C4 neurons and C3/C1, which leads to very interesting ideas in the Discussion. In doing this they focus on two C3 neurons that give similar results. Although they have clearly identified the C3 neurons correctly in 7D-E, I do not believe that they have identified those two C3 cells correctly in Figure 7I-J and supplemental 7-1A,B where the entire cluster is labeled.

The authors should do additional experiments using pre-existing and previously validated reagents. (We cannot assume any primacy of one study over another), nevertheless, it is necessary to compare carefully their results to the pre-existing studies – they should use foxO lof alleles, and the existing anti-FoxO antibodies.

The authors should explicitly explain what is being compared to what in statistical tests. Some are unclear – especially the 2-way ANOVA tests. If it was necessary (and we can't tell from the way data is presented), they should explicitly spell out how they did corrections for multiple tests.

Reviewer #1:

During nutrient restriction, animals favor the growth of vital organs like brain at the expense of other organs. A previous study has demonstrated that in *Drosophila*, larval brain sparing takes place under starvation, which is mediated by a glia-derived Jeb and the Alk receptor. In this manuscript, Chun and colleagues describe an interesting intrinsic genetic program that protects postmitotic neurons in PNS and promote their growth under nutrient restriction. They first show the differential effects of NR in the growth of epidermal cells and C4da sensory neurons, reproducing the previous findings by the Uemura lab (Watanabe et al., 2017). Although this part is not a new finding, their analyses are very detailed with nice images. Under NR, sensory neurons undergo preferential growth, in contrast to epidermal cells, which leads to dendritic overgrowth in the low nutrient condition. They then show that the InR-TOR pathway plays an important role in dendrite overgrowth under nutrient stress, whereas InR is dispensable in larval brain sparing (Cheng et al., 2011). The author further demonstrated that this dendrite overgrowth under NR is due to a lack of autophagy induction in C4da neurons under NR, in contrast to robust autophagy in epidermal cells. The authors further show that different autophagy levels in C4da neurons and epithelial cell might be due to differential expression of FoxO, a nutrient stress sensor. The overgrowth of C4da neurons under NR led to elevated sensitization of the animal in thermal nociceptive assays. In sum, the manuscript describes a novel concept that an intrinsic program promotes dendrite growth under NR. The experiments were well designed and rigorously conducted. The interpretation of their results is careful and precise in most of cases.

However, all the NR experiments were conducted with low yeast (low amino acid supply) and normal glucose level. The authors should at least examine the dendrite overgrowth under starvation (non-feeding condition), for example, they can age the larvae under normal food condition until the critical weight (60 h ALH) and switch them to the extreme starvation condition (removal of both amino acid and glucose), as shown in a previous study (Cheng et al., 2011) and others. Can the dendritic sparing phenotype be still observed at 96 h ALH under this starvation condition? This experiment is important and will help to rule out the possibility that dendrite hyper-arborization under the current NR condition may be caused by indirect effect of prolonged larval stages. Second, regarding the model in Figure 8 the authors proposed, as the authors used low-yeast medium contains less amino acid levels but normal glucose level. Low level of amino acid likely affects the amino acid sensing pathway via Slimfast/Rheb to modulate TOR activity, which I think is highly possible. The authors may examine the role of Slimfast/Rheb in dendrite overgrowth under low yeast medium (low amino acid). The above points need to be examined.

In sum, this is an interesting and important study, thus I support the publication of this manuscript in *eLife*, after the authors properly address the following points.

1) The dendrite hyper-arborization under the current NR condition may be due to prolonged larval stages (almost 10 days). Does dendritic sparing also take place at 96 h ALH after fasting in 60-96 h ALH?

2) The authors should examine the role of the amino acid sensing pathway Slimfast/Rheb in dendrite overgrowth under low yeast medium (low amino acid). The model in Figure 8 the author proposed may be revised if Slimfast and Rheb play an important role in the dendrite sparing under low yeast condition. If the results are not positive, the author at least need to discuss the possibility of the amino acid sensing pathway in the Discussion part.

3) Regarding the Foxo expression in epidermal cells and C4da neurons, I understand that the author did an excellent job to generate make foxo-Gal4 driver. Foxo-Gal4 reflects the level of Foxo transcripts. However, FoxO is post translationally regulated and translocate between nuclear and cytoplasm via phosphorylation and dephosphorylation. It is surprising that Foxo-GFP knockin did not show elevated levels of the protein (enriched in nuclear) under the current NR. One possibility is that the current NR condition is too mild for detecting the Foxo-GFP sensor. Can the author check its pattern change in its expression and nuclear enrichment under the extreme starvation state (non-feeding)? I also suggest the authors to make use of various published antibodies to examine the FoxO expression and translocation, for example, the antibodies from Broihier lab (Sears and Broihier, 2016) and Leopold lab (Slaidnina et al., 2009). This will strength the FoxO results, as a key intrinsic transcriptional program.

4) For Figure 7-supplement 1, the results are interesting. The author should include them in the major Figure 7 and leave the mCherry-Atg8a result in the supplement figure.

Reviewer #2:

This study from Poe and colleagues examines the property of “nervous system sparing”, which is the ability of developing animals to enable the distribution of nutrient usage between different cell types to favor brain growth over other tissues in times of nutrient restriction. This is an important question. It remains predominantly unknown which aspects of nervous system growth are spared, and the mechanisms involved in these processes.

The authors show that varying nutrient levels changes the growth rates of epidermal cells and c4da neurons. This occurs differentially between these two cell types. At low nutrient levels the neuron growth is relatively faster than the epidermal cell growth; hence the dendrites of the neurons end up covering more of the body wall. This part of the manuscript is very well described and characterized.

The authors show that the neurons are more sensitive to reductions in InR and Tor activity in low nutrient as opposed to high nutrient conditions. This is likely, however at present how the statistical analysis is done to show this is not well described. What exact data where used in each 2-way anova. Were there appropriate corrections for multiple testing if the same samples (including the controls) were used in different 2-way anova tests.

The authors show that reduction in nutrition does not lead to the upregulation of autophagy in c4da neurons, while it does in the epithelial cells. This is a key point. The data with Atg8 and Lamp are useful in this context. (The data with Mitf-GFP are not particularly useful, they can be moved to supplementary figures.)

The role of FoxO is central to this manuscript and has been extensively described in c4da neurons before. Thus, it is a responsibility of the authors to do this new analysis with a close consideration to the previous extensive work done on the role of this factor in c4da neurons by Sears and Broihier. In this present study, Poe and colleagues use two transgenic constructs to look at FoxO expression in c4da neurons; they could not see any. However, neuron intrinsic foxo loss does affect c4da neuron patterning. The mimic-derived line to look at FoxO expression and the foxo RNAi lines are very useful. Nevertheless, important well described reagents- a loss of function allele and an antibody – are available. The authors state that "endogenous FoxO is likely expressed at levels too low to directly affect dendrite growth". On the other hand, Sears and Broihier already did show FoxO expression in c4da neurons, and they did extensive analyses using a foxo loss of function allele showing that loss of foxo does affect dendrite growth. The present analysis is incomplete until we can reconcile the differences (it would be best if the authors can directly compare some results using the same reagents as Sears and Broihier). Do both reports of FoxO activity fit into the same overall model?

FoxO levels are lower in c4da neurons than in the epithelium. If these low FoxO levels are a primary driver of the nervous system sparing process, there should be a process to create this imbalance of low foxO in the neurons, what is it?

Growing larvae in low nutrient conditions sensitizes the animals to easier triggering of the rolling response by touching with a hot probe. It is not clear to me that this is specifically due to dendrite patterning changes. Over-expression of FoxO may disrupt neuron function on many levels, it seems safer to argue that this experiment shows that neuron function is more sensitive to disruption when the larvae develop in low nutrient conditions.

Reviewer #3:

The brain has a privileged position in resource allocation in the body. Under low nutrient conditions, resources are reallocated to the brain to ensure proper functioning. Whether peripheral neuronal systems are similarly subject to such organ sparing has not been determined. Poe, Xu, and colleagues provide a convincing argument that peripheral sensory structures are in fact spared under conditions of nutrient stress. This is a very strong paper because of the novelty of the findings, the elegance of the experiments, the rigor of the analysis, the depth of the insights provided, and the clarity with which complex manipulations and molecular pathways are explained.

The authors identify a case of sensory neuron organ sparing under low nutrient conditions and show that the Insulin Receptor-Tor pathway is responsible for this preferential growth. They identify greater susceptibility of epidermal cells vs. neurons to low nutrient conditions and provide strong evidence that this difference is mediated by the lack of induction of autophagy in sensory neurons under growth stress. The authors also show, interestingly, that different types of somatosensory neurons show different preferential growth under nutrient restriction. They speculate that this difference may be due to arbor growth mechanisms and coupling to the epidermis rather than nutrition availability.

In general, the authors did a very good job of quantifying phenotypes, addressing alternative explanations for their results, and verifying their conclusions using multiple approaches. My comments are mostly minor, with the first being the only one that could potentially affect some of the conclusions.

1) The authors make some comparisons between the C4 neurons and C3/C1, which leads to very interesting ideas in the Discussion. In doing this they focus on two C3 neurons that give similar results. Although they have clearly identified the C3 neurons correctly in 7D-E, I do not believe that they have identified those two C3 cells correctly in Figure 7I-J and supplemental 7-1A,B where the entire cluster is labeled. At the least, they have not labeled them in a way that would allow them to be discriminated from others in the cluster based on anything other than cell position and cell body shape. Based on those criteria their labeling is inconsistent with what I think is the likely identity. I hope the authors excuse this assertion if they are confident in their assignment. If there is a consistent mislabeling though this could affect the data gathered and conclusions that are drawn in this part of the manuscript.

---

## [Author Response]

Essential revisions:The authors make some comparisons between the C4 neurons and C3/C1, which leads to very interesting ideas in the Discussion. In doing this they focus on two C3 neurons that give similar results. Although they have clearly identified the C3 neurons correctly in 7D-E, I do not believe that they have identified those two C3 cells correctly in Figure 7I-J and supplemental 7-1A,B where the entire cluster is labeled.

We thank the reviewer very much for spotting the errors. We feel sorry for mislabeling C3da cell bodies in Figure 7I-J, supplement 7-1A and 1B. Therefore, we have followed the reviewer’s suggestion and re-examined the levels of autophagy and lysosomes in C3da neurons using C3da-specific driver *19-12*. The results are consistent with our conclusion that autophagy and lysosome levels are low in C3da neurons. The new results are included in the new Figure 7.

The authors should do additional experiments using pre-existing and previously validated reagents. (We cannot assume any primacy of one study over another), nevertheless, it is necessary to compare carefully their results to the pre-existing studies – they should use foxO lof alleles, and the existing anti-FoxO antibodies.

We appreciate the reviewer’s concern and therefore have tested published *foxo* reagents. In particular, we tested *foxo^Δ94^*, a *foxo* null allele used in a previous study (Sears and Broihier, 2016), and a rabbit anti-FoxO antibody from Leopold lab (Slaidina et al., 2009). We were unable to obtain the FoxO antibody from Broihier lab.

Although *foxo^Δ94^*was reported to be viable in two early studies, in our tests using HY, LY, and a standard cornmeal/molasses media, *foxo^Δ94^* was lethal before 3^rd^ instar. Therefore, we were unable to examine dendrite growth of C4da neurons in *foxo^Δ94^* mutant larvae. We think the discrepancy is likely due to different media being used in ours and earlier studies. We were unable to obtain information regarding the recipes used by Sears and Broihier, 2016.

We were able to test FoxO staining using the antibody from Leopold lab. The results were consistent with data obtained using FoxO-GFP. FoxO staining signals overlapped with FoxO-GFP and is reduced upon *foxo* knockdown. These results are included in the new Figure 4 and Figure 4—figure supplement 2.

The authors should explicitly explain what is being compared to what in statistical tests. Some are unclear – especially the 2-way ANOVA tests. If it was necessary (and we can't tell from the way data is presented), they should explicitly spell out how they did corrections for multiple tests.

We thank the reviewer for this suggestion. We have revised descriptions of our statistical tests for all figures, in both Materials and methods and figure legends, to clarify which groups were compared and what corrections were used. We conducted the statistical tests under the guidance of the Cornell Statistical Consulting Unit (CSCU).

Reviewer #1:[…]However, all the NR experiments were conducted with low yeast (low amino acid supply) and normal glucose level. The authors should at least examine the dendrite overgrowth under starvation (non-feeding condition), for example, they can age the larvae under normal food condition until the critical weight (60 h ALH) and switch them to the extreme starvation condition (removal of both amino acid and glucose), as shown in a previous study (Cheng et al., 2011) and others. Can the dendritic sparing phenotype be still observed at 96 h ALH under this starvation condition? This experiment is important and will help to rule out the possibility that dendrite hyper-arborization under the current NR condition may be caused by indirect effect of prolonged larval stages. Second, regarding the model in Figure 8 the authors proposed, as the authors used low-yeast medium contains less amino acid levels but normal glucose level. Low level of amino acid likely affects the amino acid sensing pathway via Slimfast/Rheb to modulate TOR activity, which I think is highly possible. The authors may examine the role of Slimfast/Rheb in dendrite overgrowth under low yeast medium (low amino acid). The above points need to be examined.

We thank the reviewer for suggesting these two interesting ideas, which we tested by conducting the suggested experiments.

First, our results show that larvae switched to starvation condition (no nutrient intake) after the critical weight is reached (84 hr AEL or 60 hr ALH) did not change in body size but significantly increased the normalized dendrite length over a 24-hr period. Therefore, it appears that da neurons can grow by mobilizing existing nutrient storage even in the absence of any nutrient intake, further supporting the idea of sparing of dendrite growth. The results are included in the new Figure 1—figure supplement 3.

Second, we found that *slimfast* (*slif*) knockdown in C4da neurons caused severe dendrite reduction in both HY and LY conditions, suggesting that the Slif amino acid transporter is required for proper dendrite growth regardless of the nutrient state. Interestingly, the dendrite reduction in LY food is greater, which suggests that dendrite growth under nutrient restriction may rely more on the availability of amino acid transporters. These results are included in the Figure 4—figure supplement 1.

On the other hand, *Rheb* knockdown in C4da neurons caused nutrient-state-dependent changes in dendrite growth – a mild dendrite increase in HY and a weak dendrite reduction in LY. We do not know whether this role of Rheb in nutrient-dependent dendrite regulation is dependent on Tor signaling. These results are also included in Figure 4—figure supplement 1. We also modified our model (Figure 8) and Discussion based on these new results.

In sum, this is an interesting and important study, thus I support the publication of this manuscript in eLife, after the authors properly address the following points.1) The dendrite hyper-arborization under the current NR condition may be due to prolonged larval stages (almost 10 days). Does dendritic sparing also take place at 96 h ALH after fasting in 60-96 h ALH?2) The authors should examine the role of the amino acid sensing pathway Slimfast/Rheb in dendrite overgrowth under low yeast medium (low amino acid). The model in Figure 8 the author proposed may be revised if Slimfast and Rheb play an important role in the dendrite sparing under low yeast condition. If the results are not positive, the author at least need to discuss the possibility of the amino acid sensing pathway in the Discussion part.

We have addressed these points as explained above.

3) Regarding the Foxo expression in epidermal cells and C4da neurons, I understand that the author did an excellent job to generate make foxo-Gal4 driver. Foxo-Gal4 reflects the level of Foxo transcripts. However, FoxO is post translationally regulated and translocate between nuclear and cytoplasm via phosphorylation and dephosphorylation. It is surprising that Foxo-GFP knockin did not show elevated levels of the protein (enriched in nuclear) under the current NR. One possibility is that the current NR condition is too mild for detecting the Foxo-GFP sensor. Can the author check its pattern change in its expression and nuclear enrichment under the extreme starvation state (non-feeding)? I also suggest the authors to make use of various published antibodies to examine the FoxO expression and translocation, for example, the antibodies from Broihier lab (Sears and Broihier, 2016) and Leopold lab (Slaidnina et al., 2009). This will strength the FoxO results, as a key intrinsic transcriptional program.

We appreciate the reviewer’s concerns and have performed the suggested experiments. First, we subjected the larvae to starvation and examined FoxO-GFP distribution. FoxO-GFP translocated to epidermal nuclei within 4 hrs, suggesting that starvation indeed poses more stress on larvae than nutrient restriction (LY condition). Second, using an FoxO antibody from Leopold lab (we were unable to obtain the antibody from Broihier lab), we confirmed that FoxO staining coincides with FoxO-GFP patterns. FoxO expression levels visualized by the anti-FoxO antibody are consistent with those measured by FoxO-GFP in both epidermal cells and neurons, in both HY and LY conditions. All of these new results are included in the new Figure 4 and Figure 4—figure supplement 2.

4) For Figure 7-supplement 1, the results are interesting. The author should include them in the major Figure 7 and leave the mCherry-Atg8a result in the supplement figure.

As suggested, we moved the results on lysosomal levels in da neurons to the main Figure 7. We think that the mCherry-Atg8a results are also relevant so we kept them in Figure 7.

Reviewer #2:This study from Poe and colleagues examines the property of “nervous system sparing”, which is the ability of developing animals to enable the distribution of nutrient usage between different cell types to favor brain growth over other tissues in times of nutrient restriction. This is an important question. It remains predominantly unknown which aspects of nervous system growth are spared, and the mechanisms involved in these processes.The authors show that varying nutrient levels changes the growth rates of epidermal cells and c4da neurons. This occurs differentially between these two cell types. At low nutrient levels the neuron growth is relatively faster than the epidermal cell growth; hence the dendrites of the neurons end up covering more of the body wall. This part of the manuscript is very well described and characterized.The authors show that the neurons are more sensitive to reductions in InR and Tor activity in low nutrient as opposed to high nutrient conditions. This is likely, however at present how the statistical analysis is done to show this is not well described. What exact data where used in each 2-way anova. Were there appropriate corrections for multiple testing if the same samples (including the controls) were used in different 2-way anova tests.

We thank the reviewer for this suggestion. We have revised descriptions of our statistical tests for all figures, in both Materials and methods and figure legends, to clarify which groups were compared and what corrections were used. We conducted the statistical tests under the guidance of the Cornell Statistical Consulting Unit (CSCU).

The authors show that reduction in nutrition does not lead to the upregulation of autophagy in c4da neurons, while it does in the epithelial cells. This is a key point. The data with Atg8 and Lamp are useful in this context. (The data with Mitf-GFP are not particularly useful, they can be moved to supplementary figures.)

As suggested, we moved Mitf-GFP data to Figure 3—figure supplement 2.

The role of FoxO is central to this manuscript and has been extensively described in c4da neurons before. Thus, it is a responsibility of the authors to do this new analysis with a close consideration to the previous extensive work done on the role of this factor in c4da neurons by Sears and Broihier. In this present study, Poe and colleagues use two transgenic constructs to look at FoxO expression in c4da neurons; they could not see any. However, neuron intrinsic foxo loss does affect c4da neuron patterning. The mimic-derived line to look at FoxO expression and the foxo RNAi lines are very useful. Nevertheless, important well described reagents- a loss of function allele and an antibody – are available. The authors state that "endogenous FoxO is likely expressed at levels too low to directly affect dendrite growth". On the other hand, Sears and Broihier already did show FoxO expression in c4da neurons, and they did extensive analyses using a foxo loss of function allele showing that loss of foxo does affect dendrite growth. The present analysis is incomplete until we can reconcile the differences (it would be best if the authors can directly compare some results using the same reagents as Sears and Broihier). Do both reports of FoxO activity fit into the same overall model?

We appreciate the reviewer’s concern and therefore have tested published *foxo* reagents. In particular, we tested *foxo^Δ94^*, a *foxo* null allele (Sears and Broihier, 2016), and a rabbit anti-FoxO antibody from Leopold lab (Slaidina et al., 2009). We were unable to obtain the FoxO antibody from Broihier lab.

Although *foxo^Δ94^*was reported to be viable in two early studies, in our tests using HY, LY, and a standard cornmeal/molasses media, *foxo^Δ94^* was lethal before 3^rd^ instar. Therefore, we were unable to examine dendrite growth of C4da neurons in *foxo^Δ94^* mutant larvae. We think the discrepancy is likely due to different media being used in ours and earlier studies. We were unable to obtain information regarding the recipes used by Sears and Broihier, 2016.

We were able to test FoxO staining using the antibody from Leopold lab. The results were consistent with data obtained using FoxO-GFP. FoxO staining signals overlapped with FoxO-GFP and is reduced upon *foxo* knockdown. These results are included in the new Figure 4 and Figure 4—figure supplement 2.

Although our results seem to be contradictory to those reported by Sears and Broihier, 2016, we should point out that our primary means of measuring relative dendrite growth (normalized dendrite length) is different from that used by the previous study (box counting). We in fact detected a mild reduction of dendrite density, which is more similar to the box counting method, when we knocked down *foxo* in C4da neurons under LY. In addition, as we demonstrated in this study, the relative dendrite growth depends tightly on the nutrient condition and the role of FoxO in dendrite growth is also nutrient-dependent. A possible reason for the discrepancy between our and the earlier studies could be due to different media used. We unfortunately could not obtain the exact information about the media used in the earlier study.

FoxO levels are lower in c4da neurons than in the epithelium. If these low FoxO levels are a primary driver of the nervous system sparing process, there should be a process to create this imbalance of low foxO in the neurons, what is it?

We do not know exactly how the imbalance of FoxO expression between epidermal cells and neurons is created. But we propose that the imbalance could be a result of the different genetic programs between epidermal cells and neurons that control many other differentially expressed genes.

Growing larvae in low nutrient conditions sensitizes the animals to easier triggering of the rolling response by touching with a hot probe. It is not clear to me that this is specifically due to dendrite patterning changes. Over-expression of FoxO may disrupt neuron function on many levels, it seems safer to argue that this experiment shows that neuron function is more sensitive to disruption when the larvae develop in low nutrient conditions.

We agree with the reviewer that this experiment does not prove a causal relationship between dendrite level and neuronal function. Therefore, we added the possibility that FoxO expression may affect neuronal function independent of its effect on dendrite growth. The suggestion that neuronal function is more sensitive to disruption in low nutrient conditions is an interesting one. However, we feel that we do not have enough evidence to draw this conclusion, because we have not examined many other means of disrupting neurons under both conditions.

Reviewer #3:[…]1) The authors make some comparisons between the C4 neurons and C3/C1, which leads to very interesting ideas in the Discussion. In doing this they focus on two C3 neurons that give similar results. Although they have clearly identified the C3 neurons correctly in 7D-E, I do not believe that they have identified those two C3 cells correctly in Figure 7I-J and supplemental 7-1A,B where the entire cluster is labeled. At the least, they have not labeled them in a way that would allow them to be discriminated from others in the cluster based on anything other than cell position and cell body shape. Based on those criteria their labeling is inconsistent with what I think is the likely identity. I hope the authors excuse this assertion if they are confident in their assignment. If there is a consistent mislabeling though this could affect the data gathered and conclusions that are drawn in this part of the manuscript.

We thank the reviewer very much for spotting the errors. We are sorry for mislabeling C3da cell bodies in Figure 7I-J, supplement 7-1A and 1B. Therefore, we have followed the reviewer’s suggestion and re-examined the levels of autophagy and lysosomes in C3da neurons using C3da-specific driver *19-12*. The results are consistent with our conclusion that autophagy and lysosome levels are low in C3da neurons. The new results are included in the new Figure 7.